# SWIFT: Rapid Decentralized Federated Learning via Wait-Free Model Communication

**Marco Bornstein, Tahseen Rabbani, Evan Wang, Amrit Singh Bedi, & Furong Huang**
Department of Computer Science, University of Maryland
{marcob, trabbani, ezw, amritbd, furongh}@umd.edu

## Abstract

The decentralized Federated Learning (FL) setting avoids the role of a potentially unreliable or untrustworthy central host by utilizing groups of clients to collaboratively train a model via localized training and model/gradient sharing. Most existing decentralized FL algorithms require synchronization of client models where the speed of synchronization depends upon the slowest client. In this work, we propose SWIFT: a novel wait-free decentralized FL algorithm that allows clients to conduct training at their own speed. Theoretically, we prove that SWIFT matches the gold-standard iteration convergence rate $\mathcal{O}(1/\sqrt{T})$ of parallel stochastic gradient descent for convex and non-convex smooth optimization (total iterations $T$). Furthermore, we provide theoretical results for IID and non-IID settings *without* any bounded-delay assumption for slow clients which is required by other asynchronous decentralized FL algorithms. Although SWIFT achieves the same iteration convergence rate with respect to $T$ as other state-of-the-art (SOTA) parallel stochastic algorithms, it converges faster with respect to run-time due to its wait-free structure. Our experimental results demonstrate that SWIFT's run-time is reduced due to a large reduction in communication time per epoch, which falls by an order of magnitude compared to synchronous counterparts. Furthermore, SWIFT produces loss levels for image classification, over IID and non-IID data settings, upwards of 50% faster than existing SOTA algorithms.

## 1 Introduction

Federated Learning (FL) is an increasingly popular setting to train powerful deep neural networks with data derived from an assortment of clients. Recent research (Lian et al., 2017; Li et al., 2019; Wang & Joshi, 2018) has focused on constructing decentralized FL algorithms that overcome speed and scalability issues found within classical centralized FL (McMahan et al., 2017; Savazzi et al., 2020). While decentralized algorithms have eliminated a major bottleneck in the distributed setting, the central server, their scalability potential is still largely untapped. Many are plagued by high communication time per round (Wang et al., 2019). Shortening the communication time per round allows more clients to connect and then communicate with one another, thereby increasing scalability.

Due to the synchronous nature of current decentralized FL algorithms, communication time per round, and consequently run-time, is amplified by parallelization delays. These delays are caused by the slowest client in the network. To circumvent these issues, asynchronous decentralized FL algorithms have been proposed (Lian et al., 2018; Luo et al., 2020; Liu et al., 2022; Nadiradze et al., 2021). However, these algorithms still suffer from high communication time per round. Furthermore, their communication protocols either do not propagate models well throughout the network (via gossip algorithms) or require partial synchronization. Finally, these asynchronous algorithms rely on a

Workshop on Federated Learning: Recent Advances and New Challenges, in Conjunction with NeurIPS 2022 (FL-NeurIPS'22). This workshop does not have official proceedings and this paper is non-archival.

| Algorithm | Iteration Convergence Rate | Client ($i$) Comm-Time Complexity | Neighborhood Avg. | Asynchronous | Private Memory |
|---|---|---|---|---|---|
| D-SGD | $\mathcal{O}(1/\sqrt{T})$ | $\mathcal{O}(T\max_{j\in\mathcal{N}_i}\mathscr{C}_j)$ | ✓ | ✗ | ✓ |
| PA-SGD | $\mathcal{O}(1/\sqrt{T})$ | $\mathcal{O}(|\mathcal{C}_s|\max_{j\in\mathcal{N}_i}\mathscr{C}_j)$ | ✓ | ✗ | ✓ |
| LD-SGD | $\mathcal{O}(1/\sqrt{T})$ | $\mathcal{O}(|\mathcal{C}_s|\max_{j\in\mathcal{N}_i}\mathscr{C}_j)$ | ✓ | ✗ | ✓ |
| AD-PSGD | $\mathcal{O}(\tau/\sqrt{T})$ | $\mathcal{O}(T\mathscr{C}_i)$ | ✗ | ✓ | ✗ |
| **SWIFT** | $\mathcal{O}(1/\sqrt{T})$ | $\mathcal{O}(|\mathcal{C}_s|\mathscr{C}_i)$ | ✓ | ✓ | ✓ |

**(1)** Notation: total iterations $T$, communication set $\mathcal{C}_s$ ($|\mathcal{C}_s| < T$), client $i$'s neighborhood $\mathcal{N}_i$, maximal bounded delay $\tau$, and client $i$'s communication time per round $\mathscr{C}_i$. **(2)** As compared to AD-PSGD, SWIFT does not have a $\tau$ convergence rate term due to using an expected client delay in analysis.

Table 1: Rate and complexity comparisons for decentralized FL algorithms.

deterministic bounded-delay assumption, which ensures that the slowest client in the network updates at least every $\tau$ iterations. This assumption is satisfied only under certain conditions (Abbasloo & Chao, 2020), and worsens the convergence rate by adding a sub-optimal reliance on $\tau$.

To remedy these drawbacks, we propose the **S**hared **Wa**It-**F**ree **T**ransmission (SWIFT) algorithm: an efficient, scalable, and high-performing decentralized FL algorithm. Unlike other decentralized FL algorithms, SWIFT obtains minimal communication time per round due to its wait-free structure. Furthermore, SWIFT is the first asynchronous decentralized FL algorithm to obtain an optimal $\mathcal{O}(1/\sqrt{T})$ convergence rate (aligning with stochastic gradient descent) *without a bounded-delay assumption*. Instead, SWIFT leverages the expected delay of each client (detailed in our remarks within Section 6). Experiments validate SWIFT's efficiency, showcasing a reduction in communication time by nearly an order of magnitude and run-times by upwards of 35%. All the while, SWIFT remains at state-of-the-art (SOTA) global test/train loss for image classification compared to other decentralized FL algorithms. We summarize our ***main contributions*** as follows.

▷ **(1)** Propose a novel wait-free decentralized FL algorithm (called SWIFT) and prove its theoretical convergence without a bounded-delay assumption.

▷ **(2)** Implement a novel pre-processing algorithm to ensure non-symmetric and non-doubly stochastic communication matrices are symmetric and doubly-stochastic under expectation.

▷ **(3)** Provide the first theoretical client-communication error bound for non-symmetric and non-doubly stochastic communication matrices in the asynchronous setting.

▷ **(4)** Demonstrate experimentally a significant reduction in communication time and run-time per epoch for CIFAR-10 classification in both IID and non-IID settings compared to synchronous decentralized FL algorithms.

## 2  Related Works

**Asynchronous Learning.** HOGWILD! (Recht et al., 2011), AsySG-Con (Lian et al., 2015), and AD-PSGD (Lian et al., 2017) are seminal examples of asynchronous algorithms that allow clients to proceed at their own pace. However, these methods require a shared memory/oracle from which clients grab the most up-to-date global parameters (e.g. the current graph-averaged gradient). By contrast, SWIFT relies on a message passing interface (MPI) to exchange parameters between neighbors, rather than interfacing with a shared memory structure. Algorithmically, each client relies on local memory to store current neighbor weights. To circumvent local memory overload, common in IoT clusters (Li et al., 2018), clients maintain a mailbox containing neighbor models: each client pulls out neighbor models one at a time to sequentially compute the desired aggregated statistics.

**Decentralized Stochastic Gradient Descent (SGD).** The predecessor to decentralized FL is gossip learning (Boyd et al., 2006; Hegedűs et al., 2021). Gossip learning was first introduced by the control community to assist with mean estimation of decentrally-hosted data distributions (Aysal et al., 2009; Boyd et al., 2005). Now, SGD-based gossip algorithms are used to solve large-scale machine learning tasks (Lian et al., 2015, 2018; Ghadimi et al., 2016; Nedic & Ozdaglar, 2009; Recht et al., 2011; Agarwal & Duchi, 2011). A key feature of gossip learning is the presence of a globally shared oracle/memory with whom clients exchange parameters at the end of training rounds (Boyd et al., 2006). While read/write-accessible shared memory is well-suited for a single-organization ecosystem (i.e. all clients are controllable and trusted), this is unrealistic for more general edge-based paradigms. Neighborhood-based communication and aggregation algorithms, such as D-SGD (Lian et al., 2017) and PA-SGD (Li et al., 2019), can theoretically and empirically

outperform their centralized counterparts, especially under heterogeneous client data distributions. Unfortunately, these algorithms suffer from synchronization slowdowns. SWIFT is asynchronous (avoiding slowdowns), utilizes neighborhood averaging, and does not require shared memory.

**Communication Under Expectation.** Few works in FL center on communication uncertainty. In (Ye et al., 2022), a lightweight, yet unreliable, transmission protocol is constructed in lieu of slow heavyweight protocols. A synchronous algorithm is developed to converge under expectation of an unreliable communication matrix (probabilistic link reliability). SWIFT also convergences under expectation of a communication matrix, yet in a different and asynchronous setting. SWIFT is already lightweight and reliable, and our use of expectation does not regard link reliability.

**Communication Efficiency.** Minimizing each client $i$'s communication time per round $\mathscr{C}_i$ is a challenge in FL, as the radius of information exchange can be large (Kairouz et al., 2021). MATCHA (Wang et al., 2019) decomposes the base network into $m$ disjoint matchings. Every epoch, a random sub-graph is generated from a combination of matchings, each having an activation probability $p_k$. Clients then exchange parameters along this sub-graph. This requires a total communication-time complexity of $\mathcal{O}(T \sum_{k=1}^{m} p_k \max_{j \in \mathcal{N}_i} \mathscr{C}_j)$, where $\mathcal{N}_i$ are client $i$'s neighbors. LD-SGD (Li et al., 2019) and PA-SGD (Wang & Joshi, 2018) explore how reducing the number of neighborhood parameter exchanges affects convergence. Both algorithms create a communication set $\mathcal{C}_s$ (defined in Appendix C) that dictate when clients communicate with one another. The communication-time complexities are listed in Table 1. These methods, however, are synchronous and their communication-time complexities depend upon the slowest neighbor $\max_{j \in \mathcal{N}_i} \mathscr{C}_j$. SWIFT improves upon this, achieving a communication-time complexity depending on a client's own communication-time per round. Unlike AD-PSGD Lian et al. (2018), which achieves a similar communication-time complexity, SWIFT allows for periodic communication, uses only local memory, and does not require a bounded-delay assumption.

# 3   Problem Formulation

**Decentralized FL.** In the FL setting, we have $n$ clients represented as vertices of an arbitrary communication graph $\mathcal{G}$ with vertex set $\mathcal{V} = \{1, \ldots, n\}$ and edge set $\mathcal{E} \subseteq \mathcal{V} \times \mathcal{V}$. Each client $i$ communicates with one-hop neighboring clients $j$ such that $(i, j) \in \mathcal{E}$. We denote the neighborhood for client $i$ as $\mathcal{N}_i$, and clients work in tandem to find the global model parameters $x$ by solving:

$$\min_{x \in \mathbb{R}^d} f(x) := \sum_{i=1}^{n} p_i f_i(x), \quad f_i(x) := \mathbb{E}_{\xi_i \sim \mathcal{D}_i}\big[\ell(x, \xi)\big], \quad \sum_{i=1}^{n} p_i = 1, \quad p_i \geq 0. \tag{1}$$

The global objective function $f(x)$ is the weighted average of all local objective functions $f_i(x)$. In Equation 1, $p_i, \forall i \in [n]$ denotes the client influence score. This term controls the influence of client $i$ on the global consensus model, forming the *client influence vector* $p = \{p_i\}_{i=1}^{n}$. These scores also reflect the sampling probability of each client. We note that each local objective function $f_i(x)$ is the expectation of loss function $\ell$ with respect to potentially different local data $\xi_i = \{\xi_{i,j}\}_{j=1}^{M}$ from each client $i$'s distribution $\mathcal{D}_i$, i.e., $\xi_{i,j} \sim \mathcal{D}_i$. The total number of iterations is denoted as $T$.

**Existing Inter-Client Communication in Decentralized FL.** All clients balance their individual training with inter-client communications in order to achieve consensus while operating in a decentralized manner. The core idea of decentralized FL is that each client communicates with its neighbors (connected clients) and shares local information. Balancing individual training with inter-client communication ensures individual client models are well-tailored to personal data while remaining (i) robust to other client data, and (ii) able to converge to an optimal consensus model.

**Periodic Averaging.** Algorithms such as Periodic Averaging SGD (PA-SGD) (Wang & Joshi, 2018) and Local Decentralized SGD (LD-SGD) reduce communication time by performing *multiple* local updates before synchronizing. This process is accomplished through the use of a communication set $\mathcal{C}_s$, which defines the set of iterations a client must perform synchronization,

$$\mathcal{C}_s = \{t \in \mathbb{N} \mid t \bmod (s + 1) = 0, \ t \leq T\}. \tag{2}$$

We adopt this communication set notation, although synchronization is unneeded in our algorithm.

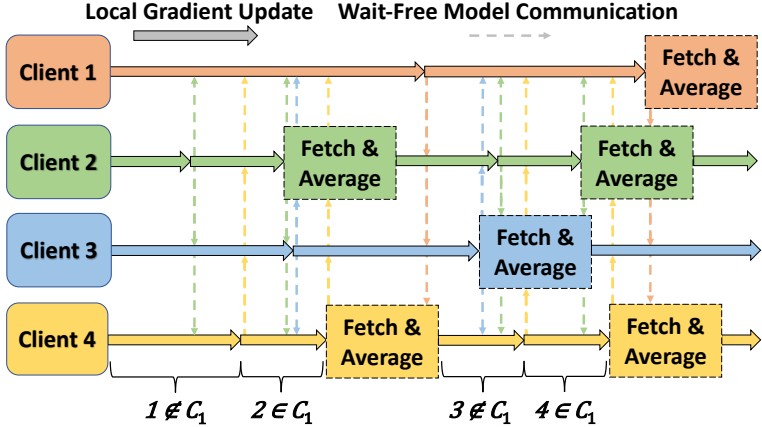

Figure 1: SWIFT schematic with $\mathcal{C}_s = \mathcal{C}_1$ (i.e., clients communicate every two local update steps).

## 4 Shared Wait-Free Transmission (SWIFT) Federated Learning

In this section, we present the **S**hared **W**a**I**t-**F**ree **T**ransmission (SWIFT) Algorithm. SWIFT is an asynchronous algorithm that allows clients to work at their own speed. Therefore, it removes the dependency on the slowest client which is the major drawback of synchronous settings. Moreover, unlike other asynchronous algorithms, SWIFT does not require a bound on the speed of the slowest client in the network and allows for neighborhood averaging and periodic communication.

**A SWIFT Overview.** Each client $i$ runs SWIFT in parallel, first receiving an initial model $x_i$, communication set $\mathcal{C}_s$, and counter $c_i \leftarrow 1$. SWIFT is concisely summarized in the following steps:
**(0)** Determine client-communication weights $w_i$ via Algorithm 2.
**(1)** Broadcast the local model to all neighboring clients.
**(2)** Sample a random local data batch of size $M$.
**(3)** Compute the gradient update of the loss function $\ell$ with the sampled local data.
**(4)** Fetch and store neighboring local models, and average them with one's own local model if $c_i \in \mathcal{C}_s$.
**(5)** Update the local model with the computed gradient update, as well as the counter $c_i \leftarrow c_i + 1$.
**(6)** Repeat steps **(1)**-**(5)** until convergence.
A diagram and algorithmic block of SWIFT are depicted in Figure 1 and Algorithm 1 respectively.

**Active Clients, Asynchronous Iterations, and the Local-Model Matrix.** Each time a client finishes a pass through steps **(1)**-**(5)**, one global iteration is performed. Thus, the global iteration $t$ is increased after the completion of *any* client's averaging and local gradient update. The client that performs the $t$-th iteration is called the active client, and is designated as $i_t$ (Line 6 of Algorithm 1). *There is only one active client per global iteration.* All other client models remain unchanged during the $t$-th iteration (Line 16 of Algorithm 1). In synchronous algorithms, the global iteration $t$ increases *only after all clients finish an update*. SWIFT, which is asynchronous, increases the global iteration $t$ after *any* client finishes an update. In our analysis, we define local-model matrix $X^t \in \mathbb{R}^{d \times n}$ as the concatenation of all local client models at iteration $t$ for ease of notation,
$$X^t := [x_1^t, \ldots, x_n^t] \in \mathbb{R}^{d \times n}. \tag{3}$$
Inspired by PA-SGD (Wang & Joshi, 2018), SWIFT handles multiple local gradient steps before averaging models amongst neighboring clients (Line 10 of Algorithm 1). Periodic averaging for SWIFT, governed by a dynamic client-communication matrix, is detailed below.

**Wait Free.** The backbone of SWIFT is its wait-free structure. Unlike any other decentralized FL algorithms, SWIFT does not require simultaneous averaging between two clients or a neighborhood of clients. Instead, each client fetches the latest models its neighbors have sent it and performs averaging with those available (Lines 11-12 of Algorithm 1). There is no pause in local training waiting for a neighboring client to finish computations or average with, making SWIFT wait-free.

**Update Rule.** SWIFT runs in parallel with all clients performing local gradient updates and model communication simultaneously. Collectively, the update rule can be written in matrix form as
$$X^{t+1} = X^t W_{i_t}^t - \gamma G(x_{i_t}^t, \xi_{i_t}^t), \tag{4}$$

---

**Algorithm 1:** **S**hared **Wa**It-**F**ree **T**ransmission (SWIFT)

**Input** : Vertex set $\mathcal{V}$, Total steps $T$, Step-size $\gamma$, Client Influence Vector $p$, Distributions of client data $\mathcal{D}_i$, Communication set $\mathcal{C}_s$, Batch size $M$, Loss function $\ell$, and Initial model $x^0$

**Output** : Consensus model $\frac{1}{n}\sum_{i=1}^{n} x_i^T$

1 Initialize each client's local update counter $c_i \leftarrow 1, \ \forall i \in \mathcal{V}$
2 Obtain each client's new communication vector $w_i^t$ using Algorithm 2
3 **for** $t = 1, \ldots, T$ **do**
4     **if** *network topology changes* **then**
5          Renew each client's communication vector $w_i^t$ using Algorithm 2
6     Randomly **select an active client** $i_t$ according to Client Influence Probability Vector $p$
7     **Broadcast active client's model** $x_{i_t}^t$ to all its neighbors $\{k \mid w_{i_t,k}^t \neq 0, k \neq i_t\}$
8     Sample a batch of active client $i_t$'s local data $\xi_{i_t}^t := \{\xi_{i_t,m}^t\}_{m=1}^M$ from distribution $\mathcal{D}_{i_t}$
9     **Compute the gradient update**: $g(x_{i_t}^t, \xi_{i_t}^t) \leftarrow \frac{1}{M}\sum_{m=1}^M \nabla \ell(x_{i_t}^t, \xi_{i_t,m}^t)$
10     **if** *current step falls in the predefined communication set, i.e., $c_{i_t} \in \mathcal{C}_s$* **then**
11         **Fetch and store the latest models** $\{x_k^t\}$ from $i_t$'s neighbors $\{k \mid w_{i_t,k}^t \neq 0, k \neq i_t\}$
12         **Model average for the active client**: $x_{i_t}^{t+1/2} \leftarrow \sum_k w_{i_t,k}^t x_k^t + w_{i_t,i_t}^t x_{i_t}^t$
13     **else**
14          Active client model remains the same: $x_{i_t}^{t+1/2} \leftarrow x_{i_t}^t$
15     **Model update for the active client**: $x_{i_t}^{t+1} \leftarrow x_{i_t}^{t+1/2} - \gamma g(x_{i_t}^t; \xi_{i_t}^t)$
16     Update other clients: $x_j^{t+1} \leftarrow x_j^t, \quad \forall j \neq i_t$
17     Update the active client's counter $c_{i_t} \leftarrow c_{i_t} + 1$

---

where $\gamma$ denotes the step size parameter and the matrix $G(x_{i_t}^t, \xi_{i_t}^t) \in \mathbb{R}^{d \times n}$ is the zero-padded gradient of the active model $x_{i_t}^t$. The entries of $G(x_{i_t}^t, \xi_{i_t}^t)$ are zero except for the $i_t$-th column, which contains the active gradient $g(x_{i_t}^t, \xi_{i_t}^t)$. Next, we describe the client-communication matrix $W_{i_t}^t$.

**Client-Communication Matrix.** The backbone of decentralized FL algorithms is the client-communication matrix $W$ (also known as the weighting matrix). To remove all forms of synchronization and to become wait-free, SWIFT relies upon a novel client-communication matrix $W_{i_t}^t$ that is neither symmetric nor doubly-stochastic, unlike other algorithms in FL (Wang & Joshi, 2018; Lian et al., 2018; Li et al., 2019; Koloskova et al., 2020). The result of a non-symmetric and non-doubly stochastic client-communication matrix, is that averaging occurs for a single active client $i_t$ and not over a pair or neighborhood of clients. This curbs superfluous communication time.

Within SWIFT, a dynamic client-communication matrix is implemented to allow for periodic averaging. We will now define the *active client-communication matrix* $W_{i_t}^t$ in SWIFT, where $i_t$ is the active client which performs the $t$-th global iteration. $W_{i_t}^t$ can be one of two forms: (1) an identity matrix $W_{i_t}^t = I_n$ if $c_{i_t} \notin \mathcal{C}_s$ or (2) a communication matrix if $c_{i_t} \in \mathcal{C}_s$ with structure,

$$W_{i_t}^t := I_n + (w_{i_t}^t - e_{i_t})e_{i_t}^\mathsf{T}, \ w_{i_t}^t := [w_{1,i_t}^t, \ldots, w_{n,i_t}^t]^\mathsf{T} \in \mathbb{R}^n, \ \sum_{j=1}^n w_{j,i} = 1, \ w_{i,i} \geq 1/n \ \forall i. \quad (5)$$

The vector $w_{i_t}^t \in \mathbb{R}^n$ denotes *the active client-communication vector* at iteration $t$, which contains the communication coefficients between client $i_t$ and all clients (including itself). The client-communication coefficients induce a weighted average of local neighboring models. We note that $w_{i_t}^t$ is often sparse because clients are connected to few other clients only in most decentralized settings.

**Novel Client-Communication Weight Selection.** While utilizing a non-symmetric and non-doubly-stochastic client-communication matrix decreases communication time, there are technical difficulties when it comes to guaranteeing the convergence. One of the novelties of our work is that we carefully design a client-communication matrix $W_{i_t}^t$ such that it is symmetric and doubly-stochastic *under expectation of all potential active clients $i_t$* and has diagonal values greater than or equal to

**Algorithm 2:** Communication Coefficient Selection (CCS)

---

**Input** : Client Influence Score (CIS) $p_i \in \mathbb{R}$, Client Degree $d_i \in \mathbb{R}$,
      Client Neighbor Set $J_i = \{\forall j : \text{client } j \text{ is a one-hop neighbor of client } i\}, \forall i$
**Output** : Client-Communication Vector $w_i \in \mathbb{R}^n, \forall i \in [n]$

**1** **for** $i = 1 : n$ *in parallel* **do**
**2**     **if** *CIS are non-uniform* **then**
**3**         | Initialize Client-Communication Vector $w_i = [w_{1,i}, w_{2,i}, \cdots, w_{n,i}] \leftarrow (1/n)\boldsymbol{e}_i$
**4**     **else**
**5**         | Initialize Client-Communication Vector $w_i = [w_{1,i}, w_{2,i}, \cdots, w_{n,i}] \leftarrow \boldsymbol{0}$
**6**     Exchange CIS and degree with all neighbors
**7**     Store *Neighbor CIS Vector* $P^J \leftarrow [\{p_j\}_{j \in J_i}] \in \mathbb{R}^{d_i}$
**8**     Construct Neighbor Subsets $J^{\mathsf{L}}, J^{\mathsf{SE}}, J^{\mathsf{E}} \subset J_i$ as subsets of $i$'s neighbors with degree larger
        than, no larger than and equal to $d_i$ respectively
**9**     **for** $\forall j \in J^{\mathsf{L}}$ **do**
**10**         | **Wait to fetch** $w_{j,i}$ from neighbor client $j$ with a degree larger than $d_i$
**11**     Determine the sum of the *total coefficients assigned (TCA)* $s_w^i \leftarrow \sum_{m=1}^n w_{m,i}$
**12**     **if** $|J^{\mathsf{SE}}| > 0$ **then**
**13**         Determine the sum of all remaining neighbors' CIS $s_p^i \leftarrow \sum_{j \in J^{\mathsf{SE}}} P_j^J$
**14**         **if** $|J^{\mathsf{E}}| > 0$ **then**
**15**             Exchange $s_w^i$ and $s_p^i$ with all neighbors $j \in J^{\mathsf{E}}$, storing all exchanged $s_w^j, s_p^j$
**16**             Store $s_w^* \leftarrow \max\{s_w^i, s_w^j\}$ and $s_p^* \leftarrow \max\{s_p^i, s_p^j\}$ $\forall j \in J^{\mathsf{E}}$
**17**             Set $w_{j,i} \leftarrow \frac{(1-s_w^*)P_j^J}{s_p^*}$ $\forall j \in J^{\mathsf{E}}$
**18**             Recompute $s_w^i = \sum_{m=1}^n w_{m,i}$ and $s_p^i = \sum_{j \in \{J^{\mathsf{SE}} \cup i\} \setminus J^{\mathsf{E}}} P_j^J$
**19**         Set $w_{j,i} \leftarrow w_{j,i} + \frac{(1-s_w^i)P_j^J}{s_p^i}$ $\forall j \in \{J^{\mathsf{SE}} \cup i\} \setminus J^{\mathsf{E}}$ for all remaining neighbors
**20**         **Send** $w_{i,j} = \frac{(1-s_w^i)P_i^J}{s_p^i}$ to all waiting neighbors $j \in J^{\mathsf{SE}} \setminus J^{\mathsf{E}}$
**21**     **else**
**22**         | $w_{i,i} = 1 - s_w^i$

---

$1/n$. Specifically, we can write

$$\mathbb{E}_{i_t}\left[W_{i_t}^t\right] = \sum_{i=1}^n p_i\left[I_n + (w_i^t - \boldsymbol{e}_i)\boldsymbol{e}_i^{\mathsf{T}}\right] = I_n + \sum_{i=1}^n p_i(w_i^t - \boldsymbol{e}_i)\boldsymbol{e}_i^{\mathsf{T}} =: \bar{W}^t, \tag{6}$$

where, we denote $\bar{W}^t$ as the *expected client-communication matrix* with the following form,

$$[\bar{W}^t]_{i,i} = 1 + p_i(w_{i,i}^t - 1), \quad \text{and} \quad [\bar{W}^t]_{i,j} = p_j w_{i,j}^t, \text{for } i \neq j. \tag{7}$$

Note that $\bar{W}^t$ is column stochastic as the entries of any column sum to one. If we ensure that $\bar{W}^t$ is symmetric, then it will become doubly-stochastic. By Equation 7, $\bar{W}^t$ becomes symmetric if,

$$p_j w_{i,j}^t = p_i w_{j,i}^t \ \forall i, j \in \mathcal{V}. \tag{8}$$

To achieve the symmetry of Equation 8, SWIFT deploys a novel pre-processing algorithm: the Communication Coefficient Selection (CCS) Algorithm. Given any client-influence vector $p_i$, CCS determines all client-communication coefficients such that Equations 5 and 8 hold for every global iteration $t$. Unlike other algorithms, CCS focuses on the *expected* client-communication matrix, ensuring its symmetry. CCS only needs to run once, before running SWIFT. In the event that the underlying network topology changes, CCS can be run again during the middle of training. Below, we detail how CCS guarantees Equations 5 and 8 to hold.

The CCS Algorithm, presented in Algorithm 2, is a waterfall method: clients receive coefficients from their larger-degree neighbors. Every client runs CCS concurrently, with the following steps:
**(1)** Receive coefficients from larger-degree neighbors. If the largest, or tied, skip to **(2)**.
**(2)** Calculate the total coefficients already assigned $s_w$ as well as the sum of the client influence

scores for the unassigned clients $s_p$.

**(3)** Assign the leftover coefficients $1 - s_w$ to the remaining unassigned neighbors (and self) in a manner proportional to each unassigned client $i$'s percentage of the leftover influence scores $p_i/s_p$.
**(4)** If tied with neighbors in degree size, ensure assigned coefficients won't sum to larger than one.
**(5)** Send coefficients to smaller-degree neighbors.

In Algorithm 2, the terms $s_w$ and $s_p$ play a pivotal role in satisfying Equations 5 and 8 respectively. Equation 8 is satisfied by assigning weights amongst client $i$ and its neighbors $j$ as follows

$$p_j \frac{(1 - s_w)p_i}{s_p} = p_i \frac{(1 - s_w)p_j}{s_p}. \tag{9}$$

Assigning weights proportionally with respect to neighboring client influence scores and total neighbors ensures higher influence clients receive higher weighting during averaging.

## 5   SWIFT **Theoretical Analysis**

**Major Message from Theoretical Analysis.** As summarized in Table 1, the efficiency and effectiveness of decentralized FL algorithms depend on both the iteration convergence rate and communication-time complexity; their product roughly approximates the total time for convergence. In this section, we will prove that SWIFT improves the SOTA convergence time of decentralized FL as it obtains SOTA iteration convergence rate (Theorem 1) and outperforms SOTA communication-time complexity.

Before presenting our theoretical results, we first detail standard assumptions (Kairouz et al., 2021) required for the analysis.

**Assumption 1** (*L*-smooth global and local objective functions). $\|\nabla f(x) - \nabla f(y)\| \leq L \|x - y\|$.

**Assumption 2** (Unbiased stochastic gradient). $\mathbb{E}_{\xi \sim \mathcal{D}_i} [\nabla \ell(x; \xi)] = \nabla f_i(x)$ *for each* $i \in \mathcal{V}$.

**Assumption 3** (Bounded inter-client gradient variance). *The variance of the stochastic gradient is bounded for any $x$ with client $i$ sampled from probability vector $p$ and local client data $\xi$ sampled from $\mathcal{D}_i$. This implies there exist constants $\sigma, \zeta \geq 0$ (where $\zeta = 0$ in IID settings) such that:* $\mathbb{E}_i \|\nabla f(x) - \nabla f_i(x)\|^2 \leq \zeta^2 \; \forall i, \forall x, \; \mathbb{E}_{\xi \sim \mathcal{D}_i} \|\nabla f_i(x) - \nabla \ell(x; \xi)\|^2 \leq \sigma^2, \forall x.$

As mentioned in Section 4, the use of a non-symmetric, non-doubly-stochastic matrix $W_{i_t}^t$ causes issues in analysis. In Appendix B, we discuss properties of stochastic matrices, including symmetric and doubly-stochastic matrices, and formally define $\rho_\nu$, a constant related to the connectivity of the network. Utilizing our symmetric and doubly-stochastic expected client-communication matrix (constructed via Algorithm 2), we reformulate Equation 4 by adding and subtracting out $\bar{W}^t$,

$$X^{t+1} = X^t \bar{W}^t + X^t (W_{i_t}^t - \bar{W}^t) - \gamma G(x_{i_t}^t, \xi_{i_t}^t). \tag{10}$$

Next, we present our first main result in Lemma 1, establishing a client-communication error bound.

**Lemma 1** (Client-Communication Error Bound). *Following Algorithm 2, the product of the difference between the expected and actual client communication matrices is bounded as follows:*

$$\mathbb{E} \sum_{j=0}^{t} \left\| G(x_{i_j}^j, \xi_{*,j}^{i_j}) \prod_{q=j+1}^{t} (W_{i_q}^q - \bar{W}^q)(\frac{\mathbf{1_n}}{n} - e_i) \right\|^2 = \mathcal{O}\Big(\frac{\sigma^2}{M} + \mathbb{E} \sum_{j=0}^{t} \|\nabla f_{i_j}(x_{i_j}^j)\|^2\Big). \tag{11}$$

**Remark.** One novelty of our work is that we are the first to bound the client-communication error in the asynchronous decentralized FL setting. The upper bound in Lemma 1 is unique to our analysis because other decentralized works do not incorporate wait-free communication (Lian et al., 2017; Li et al., 2019; Wang & Joshi, 2018; Lian et al., 2018). Now, we are ready to present our main theorem, which establishes the convergence rate of SWIFT:

**Theorem 1** (Convergence Rate of SWIFT). *Under assumptions 1, 2 and 3 (with Algorithm 2), let* $\Delta_f := f(\bar{x}^0) - f(\bar{x}^*)$, *step-size $\gamma$, total iteration $T$, and average model $\bar{x}^t$ be defined as*

$$\gamma := \frac{\sqrt{Mn^2\Delta_f}}{\sqrt{TL} + \sqrt{M}} \leq \sqrt{\frac{Mn^2\Delta_f}{TL}}, \quad T \geq 193^2 LM\Delta_f \rho_\nu^2 n^4 p_{max}^2, \quad \bar{x}^t := \frac{1}{n} \sum_{i=1}^{n} x_i^t.$$

*Then, for the output of Algorithm 1, it holds that*

$$\frac{1}{T}\sum_{t=0}^{T-1}\mathbb{E}\left\|\nabla f(\bar{x}^t)\right\|^2 \le \frac{2\sqrt{\Delta_f}}{T} + \frac{2\sqrt{L\Delta_f}\left(1 + \frac{1921}{20}\rho_\nu\left(\sigma^2 + 6\zeta^2\sqrt{M}\right)\right)}{\sqrt{TM}}. \tag{12}$$

**Iteration Convergence Rate Remarks.** **(1)** We prove that SWIFT obtains a $\mathcal{O}(1/\sqrt{T})$ iteration convergence rate, matching the optimal rate for SGD (Dekel et al., 2012; Ghadimi & Lan, 2013; Lian et al., 2017, 2018). **(2)** Unlike existing asynchronous decentralized SGD algorithms, SWIFT's iteration convergence rate does not depend on the maximal bounded delay. Instead, we bound any delays by taking the *expectation* over the active client. The probability of each client $i$ being the active client is simply its sampling probability $p_i$. We therefore assume that each client $i$ is *expected* to perform updates at its prescribed sampling probability rate $p_i$. Clients which are often delayed in practice can be dealt with by lowering their inputted sampling probability. **(3)** SWIFT converges in fewer total iterations $T$ with respect to $n$ total clients compared to other asynchronous methods (Lian et al., 2018) ($T = \Omega(n^4 p_{max}^2)$ in SWIFT versus $T = \Omega(n^4)$ in AD-PSGD).

**Communication-Time Complexity Remarks.** **(1)** Due to its asynchronous nature, SWIFT achieves a communication-time complexity that relies only on each client's own communication time per round $\mathscr{C}_i$. This improves upon synchronous decentralized SGD algorithms, which rely upon the communication time per round of the slowest neighboring client $\max_{j\in\mathcal{N}_i}\mathscr{C}_j$. **(2)** Unlike AD-PSGD (Lian et al., 2018), which also achieves a communication-time complexity reliant on $\mathscr{C}_i$, SWIFT incorporates periodic averaging which further reduces the communication complexity from $T$ rounds of communication to $|\mathcal{C}_s|$. Furthermore, SWIFT allows for entire neighborhood averaging, and not just one-to-one gossip averaging. This increases neighborhood information sharing, improving model robustness and reducing model divergence.

**Corollary 1** (Convergence under Uniform Client Influence)**.** *In the common scenario where client influences are uniform, $p_i = 1/n \ \forall i \implies p_{max} = 1/n$, SWIFT obtains convergence improvements: total iterations $T$ with respect to the number of total clients $n$ improves to $T = \Omega(n^2)$ as compared to $T = \Omega(n^4)$ for AD-PSGD under the same conditions.*

## 6 Experiments

Below, we perform image classification experiments for a range of decentralized FL algorithms Krizhevsky et al. (2009). We compare the results of SWIFT to the following decentralized baselines:
• The most common synchronous decentralized FL algorithm: D-SGD (Lian et al., 2017).
• Synchronous decentralized FL communication reduction algorithms: PA-SGD (Wang & Joshi, 2018) and LD-SGD (Li et al., 2019).
• The most prominent asynchronous decentralized FL algorithm: AD-PSGD (Lian et al., 2018).

Finer details of the experimental setup are in Appendix A. Throughout our experiments we use two network topologies: standard ring and ring of cliques (ROC). ROC-$x$C signifies a ring of cliques with $x$ clusters. The ROC topology is more reflective of a realistic network, as networks usually have pockets of connected clients. These topologies are visualized in Figures 7 and 8 respectively.

### 6.1 Baseline Comparison

To compare the performance of SWIFT to all other algorithms listed above, we reproduce an experiment within (Lian et al., 2018). With no working code for AD-PSGD to run on anything but an extreme supercomputing cluster (Section 5.1.2 of (Lian et al., 2018)), reproducing this experiment allows us to compare the relative performance of SWIFT to AD-PSGD.

Table in Figure 2 showcases that SWIFT reduces the average epoch time, relative to D-SGD, by 35% ($\mathcal{C}_0$ and $\mathcal{C}_1$). This far outpaces AD-PSGD (as well as the other synchronous algorithms), with AD-PSGD only reducing the average epoch time by 16% relative to D-SGD. Finally, Figure 2 displays how much faster SWIFT achieves optimal train and test loss values compared to other decentralized baseline algorithms. SWIFT outperforms all other baseline algorithms even without any slow-down (which we examine in Section 6.2), where wait-free algorithms like SWIFT especially shine.

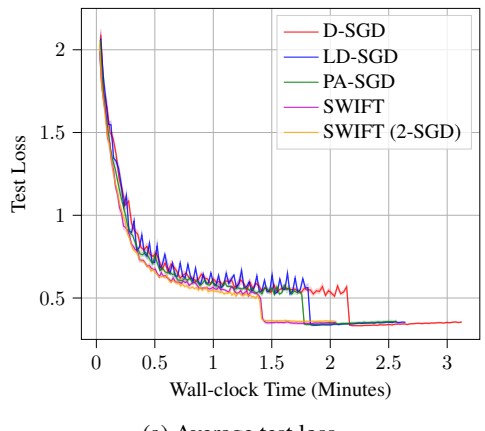
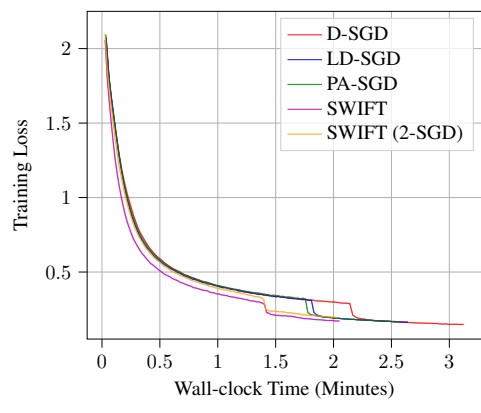

(a) Average test loss.

(b) Average train loss.

Figure 2: Baseline performance comparison on CIFAR-10 for 16 client ring.

| Decentralized FL | 16 Client Ring | | | |
|---|---|---|---|---|
| Algorithms | Epoch (s) | % Change | Comm. (s) | % Change |
| **SWIFT** ($\mathcal{C}_0$) | **1.019** | -34.60 | **0.086** | -86.28 |
| D-SGD ($\mathcal{C}_0$) | 1.558 | — | 0.627 | —- |
| AD-PSGD* ($\mathcal{C}_0$) | — | -15.86 | — | — |
| **SWIFT** ($\mathcal{C}_1$) | **1.016** | -34.79 | **0.064** | -89.79 |
| LD-SGD ($\mathcal{C}_1$) | 1.320 | -15.28 | 0.428 | -31.74 |
| PA-SGD ($\mathcal{C}_1$) | 1.281 | -17.78 | 0.358 | -42.90 |

* AD-PSGD results come from Table 4 in (Lian et al., 2018).

Table 2: Baseline average epoch and communication times on CIFAR-10 for 16 client ring.

## 6.2 Varying Heterogeneities

**Varying Degrees of Non-IIDness** Our second experiment evaluates SWIFT's efficacy at converging to a well-performing optima under varying degrees of non-IIDness. We vary the degree (percentage) of each client's data coming from one label. The remaining percentage of data is randomly sampled (IID) data over all labels. A ResNet-18 model is trained by 10 clients in a 3-cluster ROC network topology. We chose 10 clients to make the label distribution process easier: CIFAR-10 has 10 labels. As expected, when data becomes more non-IID, the test loss becomes higher and the overall accuracy lower (Table 3). We do see, however, that SWIFT converges faster, and to a lower average loss, than all other synchronous baselines (Figure 3). In fact, SWIFT with $\mathcal{C}_1$ converges *much* quicker than the synchronous algorithms. This is an important result: SWIFT converges both quicker and to a smaller loss than synchronous algorithms in the non-IID setting.

**Varying Heterogeneity of Clients** In this experiment, we investigate the performance of SWIFT under varying heterogeneity, or speed, of our clients (causing different delays). This is done with 16 clients in a ring topology. We add an artificial slowdown, suspending execution of one of the

| Decentralized FL | 10 Client Ring of Cliques - 3 Cluster Topology | |
|---|---|---|
| Algorithms | Epoch Time (s) | Communication Time (s) |
| SWIFT ($\mathcal{C}_0$) | **1.709** | **0.197** |
| D-SGD ($\mathcal{C}_0$) | 2.116 | 0.705 |
| SWIFT ($\mathcal{C}_1$) | **1.517** | **0.110** |
| LD-SGD ($\mathcal{C}_1$) | 1.973 | 0.575 |
| PA-SGD ($\mathcal{C}_1$) | 1.929 | 0.421 |

Table 3: Average epoch and communication times for non-IID setting on CIFAR-10.

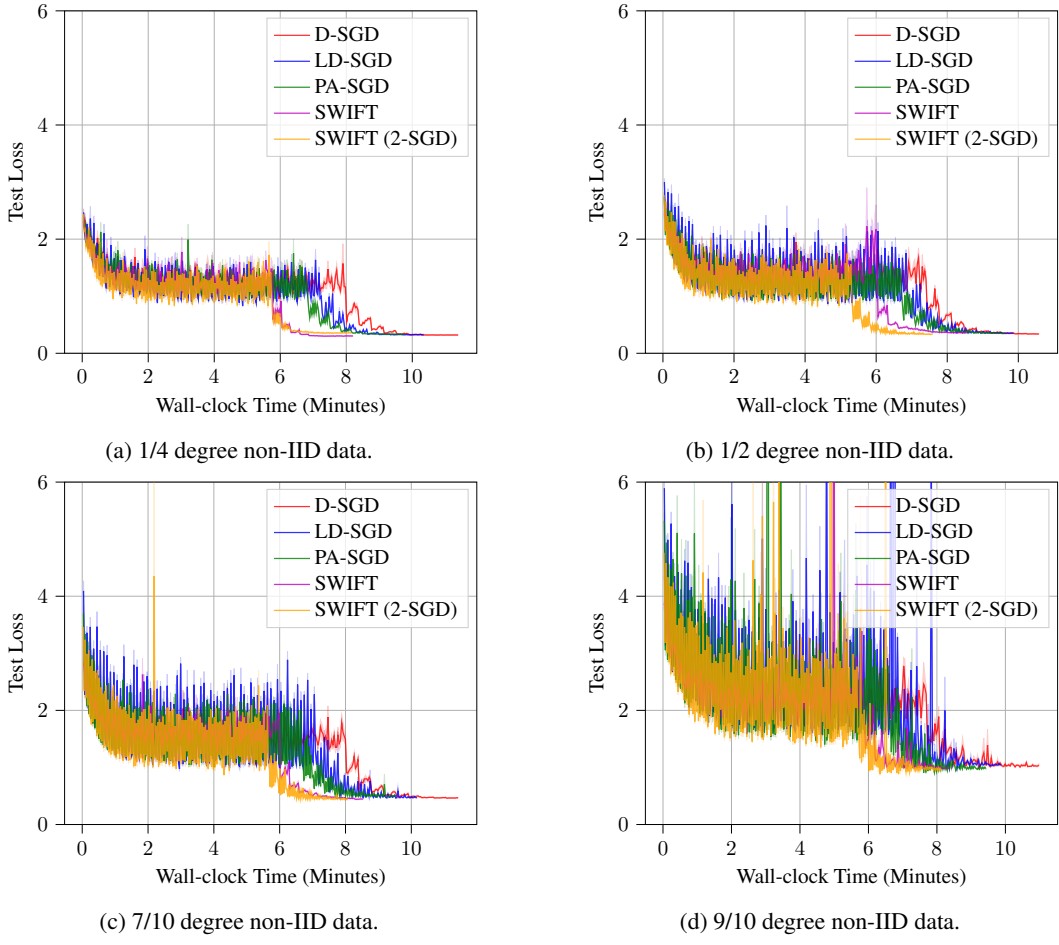

Figure 3: Average test loss for varying degrees of non-IIDness on CIFAR-10, 10 client ROC-3C.

clients such that it takes a certain amount of time longer (slowdown) to perform the computational portions of the training process. We perform tests in the case where a client is two times (2x) and four times (4x) as slow as usual. We then compare how the decentralized algorithms fare under these circumstances compared to the normal setting with no added slowdown.

| Decentralized FL | No Slowdown | | | 2x Slowdown | | | 4x Slowdown | | |
|---|---|---|---|---|---|---|---|---|---|
| Algorithms | Total (s) | Epoch (s) | Comm. (s) | Total (s) | Epoch (s) | Comm. (s) | Total (s) | Epoch (s) | Comm. (s) |
| **SWIFT** $(\mathcal{C}_0)$ | **2.69** | **1.033** | **0.087** | **2.451** | **0.964** | **0.064** | **3.054** | **1.117** | **0.091** |
| D-SGD $(\mathcal{C}_0)$ | 3.110 | 1.45 | 0.564 | 3.972 | 1.571 | 0.616 | 6.137 | 1.666 | 0.651 |
| **SWIFT** $(\mathcal{C}_1)$ | **2.44** | **0.996** | **0.061** | **2.152** | **0.928** | **0.040** | **2.847** | **1.074** | **0.065** |
| LD-SGD $(\mathcal{C}_1)$ | 3.323 | 1.353 | 0.413 | 3.762 | 1.389 | 0.428 | 5.917 | 1.412 | 0.448 |
| PA-SGD $(\mathcal{C}_1)$ | 3.183 | 1.270 | 0.331 | 3.557 | 1.262 | 0.309 | 5.743 | 1.270 | 0.318 |

Table 4: Average epoch and communication times on CIFAR-10 for 16 client ring with slowdown.

In Table 4, the average epoch, communication, and total time is displayed. Average total time includes computations, communication, and any added slowdown (wait time). SWIFT avoids large average total times as the slowdown grows larger. The wait-free structure of SWIFT allows all non-slowed clients to finish their work at their own speed. All other algorithms require clients to wait for the slowest client to finish a mini-batch before proceeding. At large slowdowns (4x), the average total time for SWIFT is nearly *half* of those for synchronous algorithms. Thus, SWIFT is very effective at reducing the run-time when clients are slow within the network.

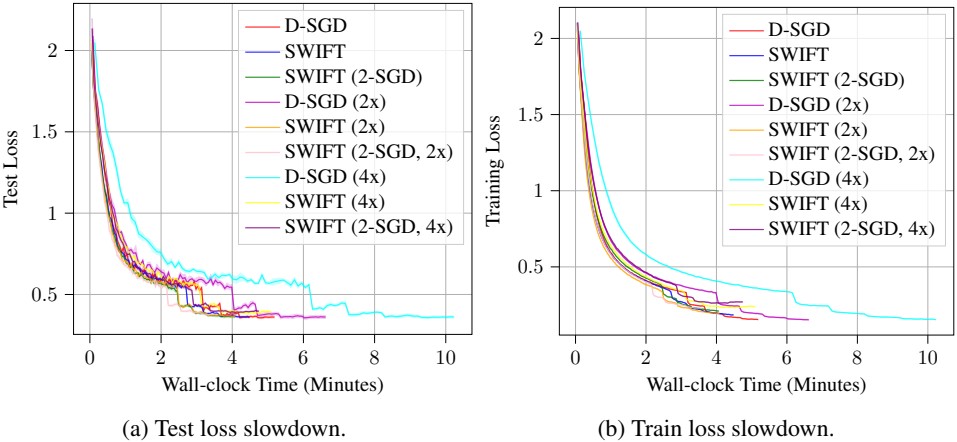

(a) Test loss slowdown.

(b) Train loss slowdown.

Figure 4: SWIFT vs. D-SGD for CIFAR-10 in 16 client ring with varying slowdown.

Figure 4 shows how SWIFT is able to converge faster to an equivalent, or smaller, test loss than D-SGD for all slowdowns. In the case of large slowdowns (4x), SWIFT significantly outperforms D-SGD, finishing in better than half the run-time. We do not include the other baseline algorithms to avoid overcrowding of the plotting space. However, SWIFT also performs much better than PA-SGD and LD-SGD as shown in Table 4. These results show that the wait-free structure of SWIFT allows it to be efficient under client slowdown.

## 6.3 Varying Numbers of Clients & Network Topologies

**Varying Numbers of Clients** In our fourth experiment, we determine how SWIFT performs versus other baseline algorithms as we vary the number of clients. In Table 5, the time per epoch for SWIFT drops by nearly the optimal factor of 2 as the number of clients is doubled. For all algorithms, there is a bit of parallel overhead when the number of clients is small, however this becomes minimal as the number of clients grow to be large (greater than 4 clients). In comparison to the synchronous algorithms, SWIFT actually *decreases* its communication time as the number of clients increases. This allows the parallel performance to be quite efficient, as shown in Figure 5.

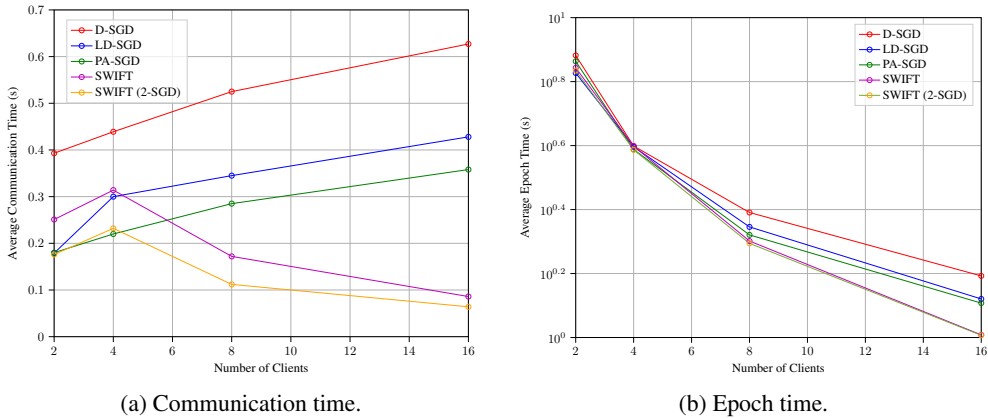

(a) Communication time.

(b) Epoch time.

Figure 5: Average communication and epoch times for increasing numbers of clients.

**Varying Topologies** Our fifth experiment analyzes the effectiveness of SWIFT versus other baseline decentralized algorithms under different, and more realistic, network topologies. In this experiment setting, we train 16 clients on varying network topologies (Table 6 and Figure 6).

| Decentralized FL | 16 Client Ring | | 8 Client Ring | | 4 Client Ring | | 2 Client Ring | |
|---|---|---|---|---|---|---|---|---|
| Algorithms | Epoch (s) | Comm. (s) | Epoch (s) | Comm. (s) | Epoch (s) | Comm. (s) | Epoch (s) | Comm. (s) |
| **SWIFT** ($\mathcal{C}_0$) | **1.019** | **0.086** | **2.003** | **0.172** | **3.964** | **0.314** | **6.971** | **0.251** |
| D-SGD ($\mathcal{C}_0$) | 1.558 | 0.627 | 2.459 | 0.525 | 3.970 | 0.439 | 7.62 | 0.393 |
| **SWIFT** ($\mathcal{C}_1$) | **1.016** | **0.064** | **1.970** | **0.112** | **3.862** | 0.232 | 6.83 | **0.176** |
| LD-SGD ($\mathcal{C}_1$) | 1.320 | 0.428 | 2.217 | 0.345 | 3.946 | 0.300 | **6.712** | 0.179 |
| PA-SGD ($\mathcal{C}_1$) | 1.281 | 0.358 | 2.093 | 0.285 | 3.871 | **0.220** | 7.303 | 0.180 |

Table 5: Average epoch and communication times on CIFAR-10 with varying clients in ring topology.

| Decentralized FL | 16 Client ROC-2C | | 16 Client ROC-4C | | 16 Client Ring | |
|---|---|---|---|---|---|---|
| Algorithms | Epoch (s) | Comm. (s) | Epoch (s) | Comm. (s) | Epoch (s) | Comm. (s) |
| **SWIFT** ($\mathcal{C}_0$) | **1.793** | **0.416** | **1.291** | **0.124** | **1.367** | **0.121** |
| D-SGD ($\mathcal{C}_0$) | 2.799 | 1.479 | 2.813 | 1.464 | 2.241 | 0.962 |
| **SWIFT** ($\mathcal{C}_1$) | **1.611** | **0.295** | **1.494** | **0.174** | **1.348** | **0.085** |
| LD-SGD ($\mathcal{C}_1$) | 2.408 | 0.987 | 2.525 | 1.105 | 2.172 | 0.517 |
| PA-SGD ($\mathcal{C}_1$) | 2.639 | 0.765 | 2.216 | 0.708 | 1.982 | 0.500 |

Table 6: Average epoch and communication times on CIFAR-10 for varying network topologies.

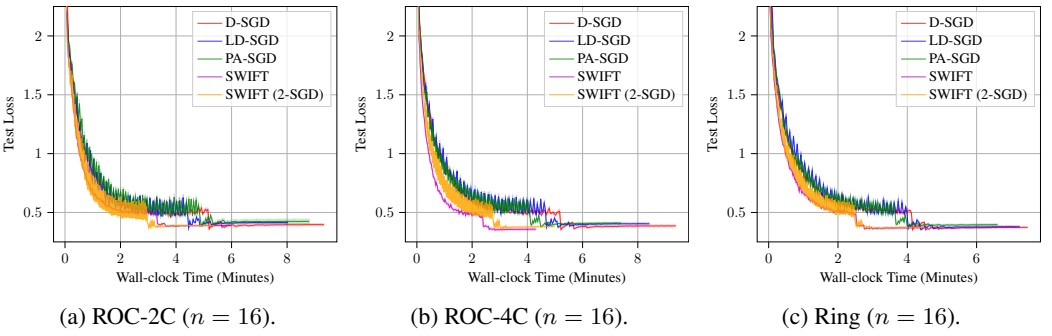

(a) ROC-2C ($n = 16$).      (b) ROC-4C ($n = 16$).      (c) Ring ($n = 16$).

Figure 6: Average test loss for varying network topologies on CIFAR-10.

## 7 Conclusion

SWIFT delivers on the promises of decentralized FL: a low communication and run-time algorithm (**scalable**) which attains SOTA loss (**high-performing**). As a wait-free algorithm, SWIFT is well-suited to rapidly solve large-scale distributed optimization problems. Empirically, SWIFT reduces communication time by almost 90% compared to baseline decentralized FL algorithms. In future work, we aim to add protocols for selecting optimal client sampling probabilities. We would like to show how varying these values can: (i) boost convergence both theoretically and empirically, and (ii) improve robustness under local client data distribution shift.

**Acknowledgments**

Bornstein, Rabbani and Huang acknowledge support by the National Science Foundation NSF-IIS-FAI program, DOD-ONR-Office of Naval Research, DOD-DARPA-Defense Advanced Research Projects Agency Guaranteeing AI Robustness against Deception (GARD), Adobe, Capital One and JP Morgan faculty fellowships. Rabbani is additionally supported by NSF DGE-1632976. Bedi acknowledges the support by Army Cooperative Agreement W911NF2120076. Bornstein additionally thanks Michael Blankenship for both helpful discussions regarding parallel code implementation and inspiration for SWIFT's name.

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

# Supplementary Material

## A  Aditional Experimental Details and Setup

### A.1  Computational Specifications

We train our consensus model on a network of nodes. Each node has an NVIDIA GeForce RTX 2080 Ti GPU. All algorithms are built in Python and communicate via Open MPI, using MPI4Py (Python bindings for MPI). The training is also done in Python, leveraging Pytorch.

### A.2  Data Partitioning

For all experiments, the training set is evenly partitioned amongst the number of clients training the consensus model. While the size of each client's partition is equal, we perform testing with data that is both (1) independent and identically distributed (iid) among all clients and (2) sorted by class and is thus non-iid. For the iid setting, each client is assigned data uniformly at random over all classes. In the non-iid setting, each client is assigned a subset of classes from which it will receive data exclusively. The $c$ classes are assigned in the following steps:
**(1)** The class subset size $n_c$, for all clients, is determined as the ceiling of the number of classes per client $n_c = \lceil \frac{c}{n} \rceil$. Each class $k$ within the class subset will take up $1/n_c$ of the client's total data partition if possible.
**(2)** The classes within each client's class subset are assigned cyclically, starting with the first client. The first client selects the first $n_c$ classes, the second client selects the next $n_c$ classes, and so on. Classes can, in some cases, be assigned to multiple clients. If the final class has been assigned, and more clients have yet to be assigned any classes, classes can be re-assigned starting at the first class.
**(3)** Each client is assigned data from the classes in their class subset cyclically ($1/n_c$ of its partition for each class), starting with the first client. If no more data is available from a specific class, the required data to fill its fraction of the partition is replaced by data from the next class.

Since we follow this data partitioning process within our experiments, each client is assigned equal partitions of data. Therefore, following the works (Lian et al., 2018; Wang et al., 2019; Ye et al., 2022; Li et al., 2019), we set the client influence scores to be uniform for all clients $p_i = 1/n \ \forall i \in \mathcal{V}$.

### A.3  Experimental Setup

Below we provide information into the hyperparameters we select for our experiments in Section 6.

| Experiment Type | Model | Epochs $E$ | $\gamma$ | $\gamma$ Decay (Rate, $E$, Freq.) | $M$ | Weight Decay | Momentum |
|---|---|---|---|---|---|---|---|
| Baseline | ResNet-18 | 200 | 0.1 | (1/10, 81 & 122, Single) | 32 | $10^{-4}$ | 0.9 |
| Vary non-IIDness | ResNet-18 | 300 | 0.8 | (1/2, 200, 10) | 32 | $10^{-4}$ | 0.9 |
| Vary Heterogeneity | ResNet-18 | 100 | 0.1 | (1/2, 50, 10) | 32 | $10^{-4}$ | 0.9 |
| Vary # of Clients | ResNet-18 | 200 | 0.1 | (1/10, 81 & 122, Single) | 32 | $10^{-4}$ | 0.9 |
| Vary Topology | ResNet-50 | 200 | 0.1 | (1/2, 100, 10) | 64 | $10^{-4}$ | 0.9 |

Table 7: Hyperparameters for all experiments.

In Table 7, one can see that the step-size decay column is split into the following sections: rate, $E$, and frequency. The rate is the decay rate for the step-size. For example, in the Baseline row, the step-size decays by 1/10. The term $E$ is the epoch at which decay begins during training. For example, in the Baseline row, the step-size decays at $E = 81$ and 122. Frequency simply is how often the step-size decays. For example, in the Vary Topology row, the step-size decays every 10 epochs.

### A.4  Network Topologies

**Ring Topology.** Please refer to Figure 7.

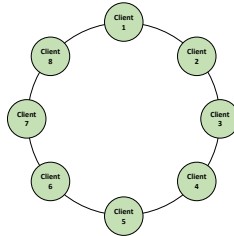

Figure 7: An 8 Client Ring.

**Ring of Cliques Topology.** Please refer to Figure 8.

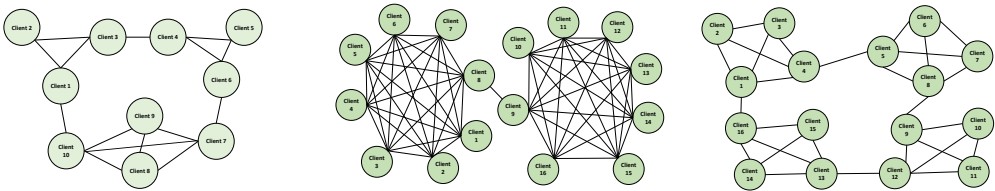

Figure 8: **(Left)** 10 Client, 3-Cluster. **(Middle)** 16 Client, 2-Cluster. **(Right)** 16 Client, 4-Cluster.

Like the works of (Jeon et al., 2021; Bellet et al., 2021), we believe that the ring of cliques network topology is a realistic topology in the decentralized setting. In many real-world setting, like one's home (smart appliances, smart speakers/displays, phones, etc.), devices are connected together in a small clique. Only a small amount of these devices have connections to other devices outside the cluster (like a phone or router). We wanted to utilize this network topology due to this realistic nature. In fact, (Bellet et al., 2021) shows that a ring of clique topology can be used in the decentralized setting to reduce the impact of label distribution skew.

## A.5 Notation Table

| Definition | Notation | ‖ | Definition | Notation |
|---|---|---|---|---|
| Global Iteration | $t$ | | Number of Clients | $n$ |
| Active Client at $t$ | $i_t$ | | Parameter Dimension | $d$ |
| Client $(i,j)$ Communication Coefficient at $t$ | $w_{i,j}^t$ | | Data Dimension | $f$ |
| Local Model of Client $i$ at $t$ | $x_i^t \in \mathbb{R}^d$ | | Step-Size | $\gamma$ |
| Local Model of Active Client at $t$ | $x_{i_t}^t \in \mathbb{R}^d$ | | Total SWIFT Iterations | $T$ |
| Mini-Batch Data from Active Client at $t$ | $\xi_{i_t}^t \in \mathbb{R}^f$ | | Mini-Batch Size (Uniform) | $M$ |
| Gradient of the Active Model at $t$ | $g(x_{i_t}^t, \xi_{i_t}^t) \in \mathbb{R}^d$ | | Communication Set | $\mathcal{C}_s$ |
| Client $i$ Communication Vector at $t$ | $w_i^t \in \mathbb{R}^n$ | | Global Data Distribution | $\mathcal{D}$ |
| Local Model Matrix at $t$ | $X^t \in \mathbb{R}^{d \times n}$ | | Global Objective | $f(x)$ |
| Zero-Padded Gradient of the Active Model at $t$ | $G(x_{i_t}^t, \xi_{i_t}^t) \in \mathbb{R}^{d \times n}$ | | Client Influence Score | $p_i$ |
| Expected Local Model Gradients at $t$ | $\bar{G}(X^t, \xi^t) \in \mathbb{R}^{d \times n}$ | | Client Influence Vector | $p \in \mathbb{R}^n$ |
| Active Client Communication Matrix at $t$ | $W_{i_t}^t \in \mathbb{R}^{n \times n}$ | | One-Hot Vector | $e_i \in \mathbb{R}^n$ |
| Expected Client Communication Matrix at $t$ | $\bar{W}^t \in \mathbb{R}^{n \times n}$ | | Identity Matrix | $I_n \in \mathbb{R}^{n \times n}$ |

Table 8: Notation Table

## B  Additional Properties and Algorithm Details

**Stochastic Matrices.**  Within our work, we use a non-symmetric, non-doubly-stochastic matrix $W_{i_t}^t$ for client averaging. Utilizing $W_{i_t}^t$ comes with some analysis issues (it is non-symmetric), however it provides the wait-free nature of SWIFT. Interestingly, $W_{i_t}^t$ does have some unique properties: it is column-stochastic. Lemma 3 proves that the product of stochastic matrices converges exponentially to a stochastic vector with common ratio $\nu \in [0, 1)$.

**Symmetric and Doubly-Stochastcic Matrices.** As mentioned in Section 5, we utilize Algorithm 2 to select client weights such that we have a symmetric and doubly-stochastic communication matrix $\bar{W}^t$ under expectation. By Lemma 2, there exists a scalar $\rho \in [0,1)$ such that $\max\{|\lambda_2((\bar{W}^t)^\intercal \bar{W}^t)|, |\lambda_n((\bar{W}^t)^\intercal \bar{W}^t)|\} \leq \rho, \forall t$. This parameter $\rho$ reflects the connectivity of the underlying graph topology. The value of $\rho$ is inversely proportionate to how fast information spreads in the client network. A small value of $\rho$ results in information spreading faster ($\rho = 0$ in centralized settings).

Within our analysis, we denote the parameter $\rho_\nu$ as a combination of $\rho$ and $\nu$:

$$\rho_\nu := \frac{n-1}{n}\left(\frac{7}{2(1-\rho)} + \frac{\sqrt{\rho}}{(1-\sqrt{\rho})^2} + \frac{384}{(1-\nu^2)}\right) \tag{13}$$

**Optimal Step-Size Under Uniform Client Influence**. The defined step-size $\gamma$ and total iterations $T$ for SWIFT is

$$\gamma := \frac{\sqrt{Mn^2\Delta_f}}{\sqrt{TL} + \sqrt{M}} \leq \sqrt{\frac{Mn^2\Delta_f}{TL}}, \quad T \geq 193^2 LM\Delta_f \rho_\nu^2 n^4 p_{max}^2.$$

Therefore, $\gamma$ can be rewritten as

$$\gamma \leq \sqrt{\frac{Mn^2\Delta_f}{(193^2 LM\Delta_f \rho_\nu^2 n^4 p_{max}^2)L}} = \sqrt{\frac{1}{193^2 L^2 \rho_\nu^2 n^2 p_{max}^2}}.$$

When the client influence scores are uniform (i.e, $p_i = 1/n \ \forall i \in \mathcal{V}$), one can see that our step-size becomes

$$\gamma = \mathcal{O}\left(\frac{1}{L}\right).$$

This mirrors the optimal step-size in analysis of gradient descent convergence to first-order stationary points $\mathcal{O}(1/L)$ (Nesterov, 1998).

**Communication Coefficient Selection (CCS) Initialization**. We include different client-communication vector initializations if the client influence scores are uniform versus non-uniform. The reason for this is to ensure that the self-weight for each client $i$, $w_{i,i}$, has a value greater than $1/n$. This naturally occurs when the CIS are uniform, however is not so when they are non-uniform.

## C Review of Existing Inter-Client Communication in Decentralized FL

**Decentralized SGD (D-SGD) (Lian et al., 2017)** One of the foundational decentralized Federated Learning algorithms is Decentralized SGD. In order to minimize Equation 1, D-SGD orchestrates a local gradient step for all clients before performing synchronous neighborhood averaging. The D-SGD process for a single client $i$ is defined as:

$$x_i^{t+1} = \sum_{j=1}^n W_{ij}\left[x_j^t - g(x_j^t, \xi_j^t)\right]. \tag{14}$$

The term $g(x_j^t, \xi_j^t)$ denotes the stochastic gradient of $x_j^t$ with mini-batch data $\xi_j^t$ sampled from the local data distribution of client $j$. The matrix $W$ is a weighting matrix, where $W_{ij}$ is the amount of $x_i^{t+1}$ which will be made up of client $j$'s local model after one local gradient step (e.g. if $W_{ij} = 1/2$, then half of $x_i^{t+1}$ will have been composed of client $j$'s model after its local gradient step). The weighting matrix only has a zero value $W_{ij} = 0$ if clients $i$ and $j$ are not connected (they are not within the same neighborhood). The values of $W_{ij}$ are generally selected ahead of time by a central host, with the usual weighting scheme being uniform. In D-SGD, model communication occurs only after all local gradient updates are finished. These gradient updates are computed in parallel.

**Periodic Averaging SGD (PA-SGD) (Wang & Joshi, 2018)** The Periodic Averaging SGD algorithm is an extension of D-SGD. In order to save communication costs when the number of clients grows to be large, PA-SGD performs model averaging after an additional $I_1$ local gradient steps. Thus, the communication set for PA-SGD is defined as:

$$\mathcal{C}_{I_1} = \{t \in \mathbb{N}|\ t \mod (I_1 + 1) = 0\}.$$

The special case of $I_1 = 0$ reduces to D-SGD. The PA-SGD process for a single client $i$ is defined as:

$$x_i^{t+1} = \begin{cases} \sum_{j=1}^w W_{ij}[x_j^t - g(x_j^t, \xi_j^t)], & t \in \mathcal{C}_{I_1} \\ x_i^t - g(x_i^t, \xi_i^t), & \text{otherwise.} \end{cases} \quad (15)$$

Compared with D-SGD, PA-SGD still suffers from the inefficiency of having to wait for the slowest client for each update. However, PA-SGD saves communication costs by reducing the frequency of communication.

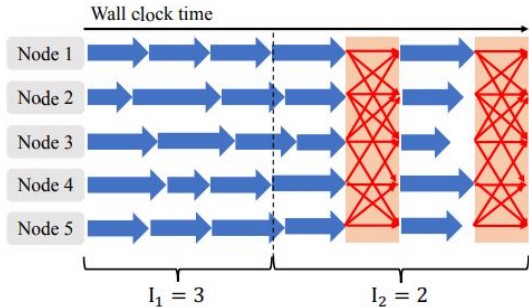

Figure 9: $I_1, I_2$ Depiction (from (Li et al., 2019)).

**Local Decentralized SGD (LD-SGD)** (Li et al., 2019)   Continuing to generalize the foundational decentralized Federated Learning algorithms is Local Decentralized SGD. LD-SGD generalizes PA-SGD by allowing multiple chunks of singular D-SGD updates, as described in Equation 14, in between the increased local gradient steps seen in PA-SGD. The number of D-SGD chunks is dictated by a new parameter $I_2$.

In this case, the communication set for LD-SGD is defined as

$$\mathcal{C}_{I_1, I_2} = \begin{cases} \bigcup \sum_{i=I_1}^{I_1+I_2} \{t \in \mathbb{N} | \ t \mod (i+1) = 0\} & \text{if } I_1 > 0, \\ \{t \in \mathbb{N}\} & \text{if } I_1 = 0. \end{cases}$$

For example, in the case $I_1 = 3, I_2 = 2$, LD-SGD will take three local gradient steps and then perform two D-SGD updates (which consists of a local gradient step and then averaging) as shown in Figure 9. The special case of $I_2 = 1$ reduces to PA-SGD. The LD-SGD process for a single client $i$ is defined as:

$$x_i^{t+1} = \begin{cases} \text{Perform } \sum_{j=1}^w W_{ij}[x_j^t - g(x_j^t, \xi_j^t)], & t \in \mathcal{C}_{I_1, I_2} \\ x_i^t - g(x_i^t, \xi_i^t), & \text{otherwise} \end{cases} \quad (16)$$

# D   Proof of the Main Theorem

Before beginning, we quickly define the expected gradient $\mathbb{E}_{i_t}\left[G(x_{i_t}^t, \xi_{*,t}^{i_t})\right]$ as

$$\bar{G}(X^t, \xi^t) := \mathbb{E}_{i_t}\left[G(x_{i_t}^t, \xi_{i_t}^t)\right] = \sum_{i=1}^n p_i G(x_{i_t}^t, \xi_{i_t}^t). \quad (17)$$

*Proof of Theorem 1.* In this theorem, we characterize the convergence of the average of *all* local models. Using the Gradient Lipschitz assumption with Equation 10 yields

$$f\left(\frac{X^{t+1}\mathbf{1_n}}{n}\right) \leq f\left(\frac{X^t\mathbf{1_n}}{n}\right) + \left\langle \nabla f\left(\frac{X^t\mathbf{1_n}}{n}\right), \frac{X^t(W_{i_t}^t - \bar{W}^t)\mathbf{1_n}}{n} - \gamma\frac{G(x_{i_t}^t, \xi_{*,t}^{i_t})\mathbf{1_n}}{n}\right\rangle$$
$$+ \frac{L}{2}\left\|\frac{X^t(W_{i_t}^t - \bar{W}^t)\mathbf{1_n}}{n} - \gamma\frac{G(x_{i_t}^t, \xi_{*,t}^{i_t})\mathbf{1_n}}{n}\right\|^2. \quad (18)$$

We first denote the average over all local models as $\bar{x}^t := \frac{X^t \mathbf{1_n}}{n}$. Taking the expectation with respect to the updating client $i_t$ yields

$$
\begin{aligned}
f(\bar{x}^{t+1}) \leq &f(\bar{x}^t) + \left\langle \nabla f(\bar{x}^t), X^t(\bar{W}^t - \bar{W}^t)\frac{\mathbf{1_n}}{n} - \gamma \bar{G}(X^t, \xi_{*,t})\frac{\mathbf{1_n}}{n} \right\rangle \\
&+ \frac{L}{2}\mathbb{E}_{i_t} \left\| X^t(W_{i_t}^t - \bar{W}^t)\frac{\mathbf{1_n}}{n} - \gamma G(x_{i_t}^t, \xi_{*,t}^{i_t})\frac{\mathbf{1_n}}{n} \right\|^2
\end{aligned}
\tag{19}
$$

$$
\begin{aligned}
= &f(\bar{x}^t) + \left\langle \nabla f(\bar{x}^t), -\gamma \bar{G}(X^t, \xi_{*,t})\frac{\mathbf{1_n}}{n} \right\rangle \\
&+ \frac{L}{2}\mathbb{E}_{i_t} \left\| X^t(W_{i_t}^t - \bar{W}^t)\frac{\mathbf{1_n}}{n} - \gamma G(x_{i_t}^t, \xi_{*,t}^{i_t})\frac{\mathbf{1_n}}{n} \right\|^2
\end{aligned}
\tag{20}
$$

$$
\begin{aligned}
= &f(\bar{x}^t) + \left\langle \nabla f(\bar{x}^t), -\frac{\gamma}{Mn} \sum_{i=1}^{n} \sum_{m=1}^{M} p_i \nabla \ell(x_i^t, \xi_{m,t}^i) \right\rangle \\
&+ \frac{L}{2}\mathbb{E}_{i_t} \left\| X^t(W_{i_t}^t - \bar{W}^t)\frac{\mathbf{1_n}}{n} - \gamma G(x_{i_t}^t, \xi_{*,t}^{i_t})\frac{\mathbf{1_n}}{n} \right\|^2
\end{aligned}
\tag{21}
$$

$$
\begin{aligned}
= &f(\bar{x}^t) - \gamma \left\langle \nabla f(\bar{x}^t), \frac{1}{Mn} \sum_{i=1}^{n} \sum_{m=1}^{M} p_i \nabla \ell(x_i^t, \xi_{m,t}^i) \right\rangle \\
&+ \frac{L}{2}\mathbb{E}_{i_t} \left\| X^t(W_{i_t}^t - \bar{W}^t)\frac{\mathbf{1_n}}{n} - \gamma G(x_{i_t}^t, \xi_{*,t}^{i_t})\frac{\mathbf{1_n}}{n} \right\|^2.
\end{aligned}
\tag{22}
$$

Taking the expectation over all local data $\mathbb{E}_{\xi \sim \mathcal{D}_i}$ yields

$$
\begin{aligned}
f(\bar{x}^{t+1}) - f(\bar{x}^t) \leq &- \frac{\gamma}{n} \left\langle \nabla f(\bar{x}^t), \sum_{i=1}^{n} p_i \nabla f_i(x_i^t) \right\rangle \\
&+ \frac{L}{2}\mathbb{E}_{\xi \sim \mathcal{D}_i, i_t} \left\| X^t(W_{i_t}^t - \bar{W}^t)\frac{\mathbf{1_n}}{n} - \gamma G(x_{i_t}^t, \xi_{*,t}^{i_t})\frac{\mathbf{1_n}}{n} \right\|^2.
\end{aligned}
\tag{23}
$$

By properties of the inner product

$$
\begin{aligned}
f(\bar{x}^{t+1}) - f(\bar{x}^t) \leq &- \frac{\gamma}{2n} \left( \|\nabla f(\bar{x}^t)\|^2 + \left\| \sum_{i=1}^{n} p_i \nabla f_i(x_i^t) \right\|^2 - \left\| \nabla f(\bar{x}^t) - \sum_{i=1}^{n} p_i \nabla f_i(x_i^t) \right\|^2 \right) \\
&+ \frac{L}{2} \underbrace{\mathbb{E}_{\xi \sim \mathcal{D}_i, i_t} \left\| X^t(W_{i_t}^t - \bar{W}^t)\frac{\mathbf{1_n}}{n} - \gamma G(x_{i_t}^t, \xi_{*,t}^{i_t})\frac{\mathbf{1_n}}{n} \right\|^2}_{:=A}.
\end{aligned}
\tag{24}
$$

**Bounding Term $A$:** Given the update equation, term $A$ can be transformed into

$$
\mathbb{E}_{\xi \sim \mathcal{D}_i, i_t} \left\| \left( X^t(W_{i_t}^t - \bar{W}^t) - \gamma G(x_{i_t}^t, \xi_{*,t}^{i_t}) \right)\frac{\mathbf{1_n}}{n} \right\|^2 = \mathbb{E}_{\xi \sim \mathcal{D}_i, i_t} \left\| \left( X^{t+1} - X^t \bar{W}^t \right)\frac{\mathbf{1_n}}{n} \right\|^2.
\tag{25}
$$

Due to the symmetric and doubly stochastic property of $\bar{W}^t$ this reduces to

$$
\mathbb{E}_{\xi \sim \mathcal{D}_i, i_t} \left\| \bar{x}^{t+1} - \bar{x}^t \right\|^2 = \mathbb{E}_{\xi \sim \mathcal{D}_i, i_t} \left\| \frac{1}{n} \sum_{i=1}^{n} \left( x_i^{t+1} - x_i^t \right) \right\|^2 = \mathbb{E}_{\xi \sim \mathcal{D}_i, i_t} \left\| \frac{1}{n} \left( x_{i_t}^{t+1} - x_{i_t}^t \right) \right\|^2
\tag{26}
$$

$$
= \frac{1}{n^2} \mathbb{E}_{\xi \sim \mathcal{D}_i, i_t} \left\| x_{i_t}^{t+1} - x_{i_t}^t \right\|^2
\tag{27}
$$

$$
\leq \frac{3}{n^2} \mathbb{E}_{\xi \sim \mathcal{D}_i, i_t} \left( \left\| x_{i_t}^{t+1} - \bar{x}^{t+1} \right\|^2 + \left\| \bar{x}^t - x_{i_t}^t \right\|^2 + \left\| \bar{x}^{t+1} - \bar{x}^t \right\|^2 \right).
\tag{28}
$$

Combining like terms yields

$$
(1 - \frac{3}{n^2})\mathbb{E}_{\xi \sim \mathcal{D}_i, i_t} \left\| \bar{x}^{t+1} - \bar{x}^t \right\|^2 \leq \frac{3}{n^2} \mathbb{E}_{\xi \sim \mathcal{D}_i, i_t} \left( \left\| x_{i_t}^{t+1} - \bar{x}^{t+1} \right\|^2 + \left\| \bar{x}^t - x_{i_t}^t \right\|^2 \right).
\tag{29}
$$

Since $n \geq 2$ (we assume at least 2 devices are running the algorithm) we find the following result

$$\mathbb{E}_{\xi \sim \mathcal{D}_i, i_t} \left\| \bar{x}^{t+1} - \bar{x}^t \right\|^2 \leq \frac{3}{(n^2 - 3)} \mathbb{E}_{\xi \sim \mathcal{D}_i, i_t} \left( \left\| x_{i_t}^{t+1} - \bar{x}^{t+1} \right\|^2 + \left\| \bar{x}^t - x_{i_t}^t \right\|^2 \right)$$

$$= \frac{3}{(n^2 - 3)} \mathbb{E}_{\xi \sim \mathcal{D}_i} \sum_{i=1}^n p_i \left( \left\| \bar{x}^{t+1} - x_i^{t+1} \right\|^2 + \left\| \bar{x}^t - x_i^t \right\|^2 \right) \qquad (30)$$

Thus, we have bounded Term $A$. Substituting this back into Equation 24 results in

$$\leq -\frac{\gamma}{2n} \left( \left\| \nabla f(\bar{x}^t) \right\|^2 + \left\| \sum_{i=1}^n p_i \nabla f_i(x_i^t) \right\|^2 - \left\| \nabla f(\bar{x}^t) - \sum_{i=1}^n p_i \nabla f_i(x_i^t) \right\|^2 \right) ]$$

$$+ \frac{3L}{2(n^2 - 3)} \mathbb{E}_{\xi \sim \mathcal{D}_i} \sum_{i=1}^n p_i \left( \left\| \bar{x}^{t+1} - x_i^{t+1} \right\|^2 + \left\| \bar{x}^t - x_i^t \right\|^2 \right). \qquad (31)$$

Taking the sum from $t = 0$ to $t = T - 1$ yields

$$f(\bar{x}^T) - f(\bar{x}^0) \leq -\frac{\gamma}{2n} \left( \sum_{t=0}^{T-1} \left\| \nabla f(\bar{x}^t) \right\|^2 - \sum_{t=0}^{T-1} \left\| \nabla f(\bar{x}^t) - \sum_{i=1}^n p_i \nabla f_i(x_i^t) \right\|^2 \right.$$

$$\left. + \sum_{t=0}^{T-1} \left\| \sum_{i=1}^n p_i \nabla f_i(x_i^t) \right\|^2 \right)$$

$$+ \frac{3L}{2(n^2 - 3)} \mathbb{E}_{\xi \sim \mathcal{D}_i} \sum_{t=0}^{T-1} \sum_{i=1}^n p_i \left( \left\| \bar{x}^{t+1} - x_i^{t+1} \right\|^2 + \left\| \bar{x}^t - x_i^t \right\|^2 \right). \qquad (32)$$

Using the Lipschitz Gradient assumption, the following term is bounded as

$$\sum_{t=0}^{T-1} \left\| \nabla f(\bar{x}^t) - \sum_{i=1}^n p_i \nabla f_i(x_i^t) \right\|^2 = \sum_{t=0}^{T-1} \left\| \sum_{i=1}^n p_i \nabla f_i(\bar{x}^t) - \sum_{i=1}^n p_i \nabla f_i(x_i^t) \right\|^2 \qquad (33)$$

$$\leq \sum_{t=0}^{T-1} \sum_{i=1}^n p_i^2 \left\| \nabla f_i(\bar{x}^t) - \nabla f_i(x_i^t) \right\|^2 \qquad (34)$$

$$\leq L^2 \sum_{t=0}^{T-1} \sum_{i=1}^n p_i^2 \left\| \bar{x}^t - x_i^t \right\|^2 \qquad (35)$$

$$\leq L^2 p_{max} \sum_{t=0}^{T-1} \sum_{i=1}^n p_i \left\| \bar{x}^t - x_i^t \right\|^2 \qquad (36)$$

Placing this back into Equation 32, and rearranging, yields

$$f(\bar{x}^T) - f(\bar{x}^0) \leq -\frac{\gamma}{2n} \sum_{t=0}^{T-1} \left\| \nabla f(\bar{x}^t) \right\|^2 - \frac{\gamma}{2n} \sum_{t=0}^{T-1} \left\| \sum_{i=1}^n p_i \nabla f_i(x_i^t) \right\|^2$$

$$+ \frac{3L}{2(n^2 - 3)} \mathbb{E}_{\xi \sim \mathcal{D}_i} \sum_{t=0}^{T-1} \sum_{i=1}^n p_i \left( \left\| \bar{x}^{t+1} - x_i^{t+1} \right\|^2 + \left\| \bar{x}^t - x_i^t \right\|^2 \right)$$

$$+ \frac{\gamma L^2 p_{max}}{2n} \sum_{t=0}^{T-1} \sum_{i=1}^n p_i \left\| \bar{x}^t - x_i^t \right\|^2 \qquad (37)$$

Given that $\bar{x}^0 = x_i^0$ for all clients $i$, one can see that

$$\sum_{t=0}^{T-1} \sum_{i=1}^n p_i \left\| \bar{x}^{t+1} - x_i^{t+1} \right\|^2 = \sum_{t=0}^{T-1} \sum_{i=1}^n p_i \left\| \bar{x}^{t+1} - x_i^{t+1} \right\|^2 + \sum_{i=1}^n p_i \left\| \bar{x}^0 - x_i^0 \right\|^2 \qquad (38)$$

$$= \sum_{t=0}^{T-1} \sum_{i=1}^n p_i \left\| \bar{x}^t - x_i^t \right\|^2 + \sum_{i=1}^n p_i \left\| \bar{x}^T - x_i^T \right\|^2 \qquad (39)$$

$$\geq \sum_{t=0}^{T-1} \sum_{i=1}^n p_i \left\| \bar{x}^t - x_i^t \right\|^2. \qquad (40)$$

Using the result of Equation 38 condenses Equation 37 into

$$f(\bar{x}^T) - f(\bar{x}^0) \leq - \frac{\gamma}{2n} \sum_{t=0}^{T-1} \left\| \nabla f(\bar{x}^t) \right\|^2 - \frac{\gamma}{2n} \sum_{t=0}^{T-1} \left\| \sum_{i=1}^{n} p_i \nabla f_i(x_i^t) \right\|^2$$

$$+ \left( \frac{3L}{(n^2 - 3)} + \frac{\gamma L^2 p_{max}}{2n} \right) \underbrace{\mathbb{E}_{\xi \sim \mathcal{D}_i} \sum_{t=0}^{T-1} \sum_{i=1}^{n} p_i \left\| \bar{x}^{t+1} - x_i^{t+1} \right\|^2}_{:=B} \quad (41)$$

**Bounding Term B.** The recursion of our update rule can be written as

$$X^{t+1} = X^0 - \gamma \sum_{j=0}^{t} G(x_{i_j}^j, \xi_{*,j}^{i_j}) \prod_{q=j+1}^{t} \bar{W}^q - \gamma \sum_{j=0}^{t} G(x_{i_j}^j, \xi_{*,j}^{i_j}) \prod_{q=j+1}^{t} (W_{i_q}^q - \bar{W}^q) \quad (42)$$

The recursion equation for the expected consensus model $\bar{x}^t$ and expected local model for client $i$ can be computed by multiplying by $\frac{\mathbf{1_n}}{n}$ and $\mathbf{e}_i$ respectively

$$\bar{x}^{t+1} = \bar{x}^0 - \gamma \sum_{j=0}^{t} G(x_{i_j}^j, \xi_{*,j}^{i_j}) \frac{\mathbf{1_n}}{n} - \gamma \sum_{j=0}^{t} G(x_{i_j}^j, \xi_{*,j}^{i_j}) \prod_{q=j+1}^{t} (W_{i_q}^q - \bar{W}^q) \frac{\mathbf{1_n}}{n} \quad (43)$$

$$x_i^{t+1} = x_i^0 - \gamma \sum_{j=0}^{t} G(x_{i_j}^j, \xi_{*,j}^{i_j}) \prod_{q=j+1}^{t} \bar{W}^q \mathbf{e}_i - \gamma \sum_{j=0}^{t} G(x_{i_j}^j, \xi_{*,j}^{i_j}) \prod_{q=j+1}^{t} (W_{i_q}^q - \bar{W}^q) \mathbf{e}_i \quad (44)$$

Using the recursive equations above transforms the bound on term B

$$\mathbb{E}_{\xi \sim \mathcal{D}_i} \sum_{t=0}^{T-1} \sum_{i=1}^{n} p_i \left\| \bar{x}^{t+1} - x_i^{t+1} \right\|^2$$

$$= \gamma^2 \mathbb{E}_{\xi \sim \mathcal{D}_i} \sum_{t=0}^{T-1} \sum_{i=1}^{n} p_i \left\| - \sum_{j=0}^{t} G(x_{i_j}^j, \xi_{*,j}^{i_j}) \left( \frac{\mathbf{1_n}}{n} - \prod_{q=j+1}^{t} \bar{W}^q \mathbf{e}_i \right) - \sum_{j=0}^{t} G(x_{i_j}^j, \xi_{*,j}^{i_j}) \prod_{q=j+1}^{t} (W_{i_q}^q - \bar{W}^q)(\frac{\mathbf{1_n}}{n} - \mathbf{e}_i) \right\|^2 \quad (45)$$

$$\leq 2\gamma^2 \mathbb{E}_{\xi \sim \mathcal{D}_i} \sum_{t=0}^{T-1} \sum_{i=1}^{n} p_i \Bigg( \left\| \sum_{j=0}^{t} G(x_{i_j}^j, \xi_{*,j}^{i_j}) \left( \frac{\mathbf{1_n}}{n} - \prod_{q=j+1}^{t} \bar{W}^q \mathbf{e}_i \right) \right\|^2$$

$$+ \left\| \sum_{j=0}^{t} G(x_{i_j}^j, \xi_{*,j}^{i_j}) \prod_{q=j+1}^{t} (W_{i_q}^q - \bar{W}^q)(\frac{\mathbf{1_n}}{n} - \mathbf{e}_i) \right\|^2 \Bigg) \quad (46)$$

$$= 2\gamma^2 \mathbb{E}_{\xi \sim \mathcal{D}_i} \sum_{t=0}^{T-1} \sum_{i=1}^{n} p_i \Bigg( \underbrace{\sum_{j=0}^{t} \left\| G(x_{i_j}^j, \xi_{*,j}^{i_j}) \left( \frac{\mathbf{1_n}}{n} - \prod_{q=j+1}^{t} \bar{W}^q \mathbf{e}_i \right) \right\|^2}_{:=B_1}$$

$$+ \underbrace{\sum_{j=0}^{t} \left\| G(x_{i_j}^j, \xi_{*,j}^{i_j}) \prod_{q=j+1}^{t} (W_{i_q}^q - \bar{W}^q)(\frac{\mathbf{1_n}}{n} - \mathbf{e}_i) \right\|^2}_{:=B_3}$$

$$+ \underbrace{2 \sum_{j=0}^{t} \sum_{j'=j+1}^{t} \langle G(x_{i_j}^j, \xi_{*,j}^{i_j})(\frac{\mathbf{1_n}}{n} - \prod_{q=j+1}^{t} \bar{W}^q \mathbf{e}_i), G(x_{i_{j'}}^{j'}, \xi_{*,j'}^{i_{j'}})(\frac{\mathbf{1_n}}{n} - \prod_{q=j'+1}^{t} \bar{W}^q \mathbf{e}_i) \rangle}_{:=B_2}$$

$$+ \underbrace{2 \sum_{j=0}^{t} \sum_{j'=j+1}^{t} \langle G(x_{i_j}^j, \xi_{*,j}^{i_j}) \prod_{q=j+1}^{t} (W_{i_q}^q - \bar{W}^q)(\frac{\mathbf{1_n}}{n} - \mathbf{e}_i), G(x_{i_{j'}}^{j'}, \xi_{*,j'}^{i_{j'}}) \prod_{q=j'+1}^{t} (W_{i_q}^q - \bar{W}^q)(\frac{\mathbf{1_n}}{n} - \mathbf{e}_i) \rangle}_{:=B_4} \Bigg) \quad (47)$$

**Bounding Term $B_1$.** Using Lemma 2,

$$\mathbb{E}_{\xi \sim \mathcal{D}_i} \sum_{j=0}^{t} \left\| G(x_{i_j}^j, \xi_{*,j}^{i_j}) \Big( \frac{\mathbf{1_n}}{n} - \prod_{q=j+1}^{t} \bar{W}^q \boldsymbol{e}_i \Big) \right\|^2$$

$$\leq \mathbb{E}_{\xi \sim \mathcal{D}_i} \sum_{j=0}^{t} \left\| G(x_{i_j}^j, \xi_{*,j}^{i_j}) \right\|^2 \left\| \Big( \frac{\mathbf{1_n}}{n} - \prod_{q=j+1}^{t} \bar{W}^q \boldsymbol{e}_i \Big) \right\|^2 \tag{48}$$

$$= \mathbb{E}_{\xi \sim \mathcal{D}_i} \sum_{j=0}^{t} \left\| \frac{1}{M} \sum_{m=1}^{M} \nabla \ell(x_{i_j}^j; \xi_{m,j}^{i_j}) \right\|^2 \left\| \Big( \frac{\mathbf{1_n}}{n} - \prod_{q=j+1}^{t} \bar{W}^q \boldsymbol{e}_i \Big) \right\|^2 \tag{49}$$

$$\leq \mathbb{E}_{\xi \sim \mathcal{D}_i} \sum_{j=0}^{t} \left\| \frac{1}{M} \sum_{m=1}^{M} \nabla \ell(x_{i_j}^j; \xi_{m,j}^{i_j}) \right\|^2 (\frac{n-1}{n}) \rho^{t-j} \tag{50}$$

Using Lemma 4, the equation above becomes

$$= 2 \sum_{j=0}^{t} \left( \frac{\sigma^2}{M} + \left\| \nabla f_{i_j}(x_{i_j}^j) \right\|^2 \right) (\frac{n-1}{n}) \rho^{t-j} \tag{51}$$

$$= 2 \left( \sum_{j=0}^{t} \frac{\sigma^2}{M} (\frac{n-1}{n}) \rho^{t-j} + \sum_{j=0}^{t} \left\| \nabla f_{i_j}(x_{i_j}^j) \right\|^2 (\frac{n-1}{n}) \rho^{t-j} \right) \tag{52}$$

$$\leq \frac{2(n-1)\sigma^2}{(1-\rho)Mn} + 2 \sum_{j=0}^{t} \left\| \nabla f_{i_j}(x_{i_j}^j) \right\|^2 (\frac{n-1}{n}) \rho^{t-j} \tag{53}$$

Taking the expectation over worker $i_j$ yields the desired bound

$$\mathbb{E}_{i_j} \left[ \frac{2(n-1)\sigma^2}{(1-\rho)Mn} + 2 \sum_{j=0}^{t} \left\| \nabla f_{i_j}(x_{i_j}^j) \right\|^2 (\frac{n-1}{n}) \rho^{t-j} \right]$$

$$= \frac{2(n-1)\sigma^2}{(1-\rho)Mn} + \frac{2(n-1)}{n} \sum_{j=0}^{t} \mathbb{E}_{i_j} \left\| \nabla f_{i_j}(x_{i_j}^j) \right\|^2 \rho^{t-j} \tag{54}$$

**Bounding Term $B_2$.** Using Lemma 2,

$$2 \sum_{j=0}^{t} \sum_{j'=j+1}^{t} \langle G(x_{i_j}^j, \xi_{*,j}^{i_j}) (\frac{\mathbf{1_n}}{n} - \prod_{q=j+1}^{t} \bar{W}^q \boldsymbol{e}_i), G(x_{i_{j'}}^{j'}, \xi_{*,j'}^{i_{j'}}) (\frac{\mathbf{1_n}}{n} - \prod_{q=j'+1}^{t} \bar{W}^q \boldsymbol{e}_i) \rangle$$

$$= 2 \sum_{j=0}^{t} \sum_{j'=j+1}^{t} \left\| G(x_{i_j}^j, \xi_{*,j}^{i_j}) \right\| \left\| (\frac{\mathbf{1_n}}{n} - \prod_{q=j+1}^{t} \bar{W}^q \boldsymbol{e}_i) \right\| \left\| G(x_{i_{j'}}^{j'}, \xi_{*,j'}^{i_{j'}}) \right\| \left\| (\frac{\mathbf{1_n}}{n} - \prod_{q=j'+1}^{t} \bar{W}^q \boldsymbol{e}_i) \right\| \tag{55}$$

For any $\alpha_{j,j'} > 0$ we find

$$\leq 2 \sum_{j=0}^{t} \sum_{j'=j+1}^{t} \left( \frac{\left\| G(x_{i_j}^j, \xi_{*,j}^{i_j}) \right\|^2 \left\| G(x_{i_{j'}}^{j'}, \xi_{*,j'}^{i_{j'}}) \right\|^2}{2\alpha_{j,j'}} \right.$$

$$\left. + \frac{\alpha_{j,j'} \left\| (\frac{\mathbf{1_n}}{n} - \prod_{q=j+1}^{t} \bar{W}^q \boldsymbol{e}_i) \right\|^2 \left\| (\frac{\mathbf{1_n}}{n} - \prod_{q=j'+1}^{t} \bar{W}^q \boldsymbol{e}_i) \right\|^2}{2} \right) \tag{56}$$

$$\leq \sum_{j \neq j'}^{t} \left( \frac{\left\| G(x_{i_j}^j, \xi_{*,j}^{i_j}) \right\|^2 \left\| G(x_{i_{j'}}^{j'}, \xi_{*,j'}^{i_{j'}}) \right\|^2}{2\alpha_{j,j'}} + \frac{\alpha_{j,j'} \rho^{t-\min\{j,j'\}}}{2} (\frac{n-1}{n})^2 \right), \alpha_{j,j'} = \alpha_{j',j} \tag{57}$$

By applying inequality of arithmetic and geometric means to the term in the last step, we can choose $\alpha_{j,j'} > 0$ s.t.

$$\leq \frac{n-1}{n} \sum_{j \neq j'}^{t} \left( \left\| G(x_{i_j}^j, \xi_{*,j}^{i_j}) \right\| \left\| G(x_{i_{j'}}^{j'}, \xi_{*,j'}^{i_{j'}}) \right\| \rho^{\frac{t - \min\{j,j'\}}{2}} \right) \tag{58}$$

$$\leq \frac{n-1}{n} \sum_{j \neq j'}^{t} \left( \frac{\left\| G(x_{i_j}^j, \xi_{*,j}^{i_j}) \right\|^2 + \left\| G(x_{i_{j'}}^{j'}, \xi_{*,j'}^{i_{j'}}) \right\|^2}{2} \rho^{\frac{t - \min\{j,j'\}}{2}} \right) \tag{59}$$

$$= \frac{n-1}{n} \sum_{j \neq j'}^{t} \left\| G(x_{i_j}^j, \xi_{*,j}^{i_j}) \right\|^2 \rho^{\frac{t - \min\{j,j'\}}{2}} \tag{60}$$

$$= \frac{n-1}{n} \sum_{j=0}^{t} \sum_{j'=j+1}^{t} \left\| G(x_{i_j}^j, \xi_{*,j}^{i_j}) \right\|^2 \rho^{\frac{t-j}{2}} \tag{61}$$

$$= \frac{n-1}{n} \sum_{j=0}^{t} \left\| G(x_{i_j}^j, \xi_{*,j}^{i_j}) \right\|^2 2(t-j)\rho^{\frac{t-j}{2}} \tag{62}$$

$$= \frac{n-1}{n} \sum_{j=0}^{t} 2(t-j)\rho^{\frac{t-j}{2}} \left\| \frac{1}{M} \sum_{m=1}^{M} \nabla\ell(x_{i_j}^j; \xi_{m,j}^{i_j}) \right\|^2 \tag{63}$$

Using Lemma 4 (and the expectation $\mathbb{E}_{\xi \sim \mathcal{D}_i}$ that was omitted above but is present) yields

$$= \frac{2(n-1)}{n} \sum_{j=0}^{t} 2(t-j)\rho^{\frac{t-j}{2}} \left( \frac{\sigma^2}{M} + \left\| \nabla f_{i_j}(x_{i_j}^j) \right\|^2 \right) \tag{64}$$

$$\leq \frac{4(n-1)\sqrt{\rho}\sigma^2}{Mn(1 - \sqrt{\rho})^2} + \frac{2(n-1)}{n} \sum_{j=0}^{t} \left\| \nabla f_{i_j}(x_{i_j}^j) \right\|^2 2(t-j)\rho^{\frac{t-j}{2}} \tag{65}$$

Taking the expectation over worker $i_j$ yields the desired bound

$$\mathbb{E}_{i_j} \left[ \frac{4(n-1)\sqrt{\rho}\sigma^2}{Mn(1 - \sqrt{\rho})^2} + \frac{2(n-1)}{n} \sum_{j=0}^{t} \left\| \nabla f_{i_j}(x_{i_j}^j) \right\|^2 2(t-j)\rho^{\frac{t-j}{2}} \right]$$

$$= \frac{4(n-1)\sqrt{\rho}\sigma^2}{Mn(1 - \sqrt{\rho})^2} + \frac{2(n-1)}{n} \sum_{j=0}^{t} \mathbb{E}_{i_j} \left\| \nabla f_{i_j}(x_{i_j}^j) \right\|^2 2(t-j)\rho^{\frac{t-j}{2}} \tag{66}$$

**Bounding Term $B_3$.**

$$\mathbb{E}_{\xi \sim \mathcal{D}_i} \sum_{j=0}^{t} \left\| G(x_{i_j}^j, \xi_{*,j}^{i_j}) \prod_{q=j+1}^{t} (W_{i_q}^q - \bar{W}^q)(\frac{\mathbf{1_n}}{n} - \boldsymbol{e}_i) \right\|^2$$

$$= \mathbb{E}_{\xi \sim \mathcal{D}_i} \sum_{j=0}^{t} \left\| G(x_{i_j}^j, \xi_{*,j}^{i_j}) \prod_{q=j+1}^{t} (W_{i_q}^q - \phi\mathbf{1}^{\mathsf{T}} + \phi\mathbf{1}^{\mathsf{T}} - \bar{W}^q)(\frac{\mathbf{1_n}}{n} - \boldsymbol{e}_i) \right\|^2 \tag{67}$$

$$= \mathbb{E}_{\xi \sim \mathcal{D}_i} \sum_{j=0}^{t} \left\| G(x_{i_j}^j, \xi_{*,j}^{i_j}) \left[ \prod_{q=j+1}^{t}(W_{i_q}^q - \phi\mathbf{1}^{\mathsf{T}})(\frac{\mathbf{1_n}}{n} - \boldsymbol{e}_i) - \prod_{q=j+1}^{t}(\bar{W}^q - \phi\mathbf{1}^{\mathsf{T}})(\frac{\mathbf{1_n}}{n} - \boldsymbol{e}_i) \right] \right\|^2 \tag{68}$$

$$\leq 2\mathbb{E}_{\xi \sim \mathcal{D}_i} \sum_{j=0}^{t} \left\| G(x_{i_j}^j, \xi_{*,j}^{i_j}) \prod_{q=j+1}^{t} (W_{i_q}^q - \phi\mathbf{1}^{\mathsf{T}})(\frac{\mathbf{1_n}}{n} - \boldsymbol{e}_i) \right\|^2$$

$$+ 2\mathbb{E}_{\xi \sim \mathcal{D}_i} \sum_{j=0}^{t} \left\| G(x_{i_j}^j, \xi_{*,j}^{i_j}) \prod_{q=j+1}^{t} (\bar{W}^q - \phi\mathbf{1}^{\mathsf{T}})(\frac{\mathbf{1_n}}{n} - \boldsymbol{e}_i) \right\|^2 \tag{69}$$

Due to the structure of $\phi 1^\mathsf{T}$, multiplying this matrix by $\frac{1_n}{n}$ or $e_i$ yields the same result. Using this, as well as the double stochasticity of $\bar{W}$, we find

$$= 2\mathbb{E}_{\xi \sim \mathcal{D}_i} \sum_{j=0}^t \left[ \left\| G(x_{i_j}^j, \xi_{*,j}^{i_j}) \prod_{q=j+1}^t (W_{i_q}^q - \phi 1^\mathsf{T})(\frac{1_n}{n} - e_i) \right\|^2 + \left\| G(x_{i_j}^j, \xi_{*,j}^{i_j}) \prod_{q=j+1}^t \bar{W}^q(\frac{1_n}{n} - e_i) \right\|^2 \right] \qquad (70)$$

$$= 2\mathbb{E}_{\xi \sim \mathcal{D}_i} \sum_{j=0}^t \left[ \left\| G(x_{i_j}^j, \xi_{*,j}^{i_j}) \prod_{q=j+1}^t (W_{i_q}^q - \phi 1^\mathsf{T})(\frac{1_n}{n} - e_i) \right\|^2 + \underbrace{\left\| G(x_{i_j}^j, \xi_{*,j}^{i_j})(\frac{1_n}{n} - \prod_{q=j+1}^t \bar{W}^q e_i) \right\|^2}_{=B_1} \right] \qquad (71)$$

Using the result of Lemma 3, as our communication graph $\mathcal{G}$ is uniformly strongly connected and $[W_{i_t}^t]_{i,i} \geq 1/n$ by construction, we see that

$$|[\prod_{q=j+1}^t W_{i_q}^q - \phi 1^\mathsf{T}]_{h,k}| \leq 4\nu^{t-j-1} \ \forall \ h, k. \qquad (72)$$

Using this result, we find that

$$|[\prod_{q=j+1}^t (W_{i_q}^q - \phi 1^\mathsf{T})(\frac{1_n}{n} - e_i)]_h| \leq 8(\frac{n-1}{n})\nu^{t-j-1} \ \forall \ h. \qquad (73)$$

We can remove the -1 exponent by doubling the constant out front

$$|[\prod_{q=j+1}^t (W_{i_q}^q - \phi 1^\mathsf{T})(\frac{1_n}{n} - e_i)]_h| \leq 16(\frac{n-1}{n})\nu^{t-j} \ \forall \ h. \qquad (74)$$

Finally, using the fact that $G(x_{i_j}^j, \xi_{*,j}^{i_j})$ is all zeros except for one column, the $i_j$-th column, yields the desired result

$$\mathbb{E}_{\xi \sim \mathcal{D}_i} \sum_{j=0}^t \left\| G(x_{i_j}^j, \xi_{*,j}^{i_j}) \prod_{q=j+1}^t (W_{i_q}^q - \phi 1^\mathsf{T})(\frac{1_n}{n} - e_i) \right\|^2$$

$$\leq \sum_{j=0}^t 256(\frac{n-1}{n})^2 \nu^{2(t-j)} \mathbb{E}_{\xi \sim \mathcal{D}_i} \left\| \frac{1}{M} \sum_{m=1}^M \nabla \ell(x_{i_j}^j, \xi_{*,j}^{i_j}) \right\|^2. \qquad (75)$$

Utilizing Lemma 4 yields

$$\leq 256(\frac{n-1}{n})^2 \sum_{j=0}^t \nu^{2(t-j)} \left( \frac{\sigma^2}{M} + \left\| \nabla f_{i_j}(x_{i_j}^t) \right\|^2 \right). \qquad (76)$$

By properties of geometric series, and taking the expectation over worker $i_j$, we find

$$\leq \frac{256\sigma^2}{(1-\nu^2)M}(\frac{n-1}{n})^2 + 256(\frac{n-1}{n})^2 \sum_{j=0}^t \mathbb{E}_{i_j} \left\| \nabla f_{i_j}(x_{i_j}^t) \right\|^2 \nu^{2(t-j)}. \qquad (77)$$

Using the bound of $B_1$ in the main proof above, we arrive at the final bound of $B_3$

$$\mathbb{E}_{\xi \sim \mathcal{D}_i} \sum_{j=0}^t \left\| G(x_{i_j}^j, \xi_{*,j}^{i_j}) \prod_{q=j+1}^t (W_{i_q}^q - \bar{W}^q)(\frac{1_n}{n} - e_i) \right\|^2$$

$$\leq \frac{512\sigma^2}{(1-\nu^2)M}(\frac{n-1}{n})^2 + \frac{4(n-1)\sigma^2}{(1-\rho)Mn} + 512(\frac{n-1}{n})^2 \sum_{j=0}^t \mathbb{E}_{i_j} \left\| \nabla f_{i_j}(x_{i_j}^t) \right\|^2 \nu^{2(t-j)}$$

$$+ \frac{4(n-1)}{n} \sum_{j=0}^t \mathbb{E}_{i_j} \left\| \nabla f_{i_j}(x_{i_j}^j) \right\|^2 \rho^{t-j} \qquad (78)$$

$$\leq \frac{4(n-1)\sigma^2}{Mn} \left( \frac{1}{(1-\rho)} + \frac{128}{(1-\nu^2)} \right)$$

$$+ \frac{4(n-1)}{n} \sum_{j=0}^t \mathbb{E}_{i_j} \left\| \nabla f_{i_j}(x_{i_j}^j) \right\|^2 \left( \rho^{t-j} + 128\nu^{2(t-j)} \right) \qquad (79)$$

**Bounding Term $B_4$.** Following similar steps as bounding Term $B_2$ we find

$$2\mathbb{E}_{\xi\sim\mathcal{D}_i}\sum_{j=0}^{t}\sum_{j'=j+1}^{t}\langle G(x_{i_j}^j,\xi_{*,j}^{i_j})\prod_{q=j+1}^{t}(W_{i_q}^q-\bar{W}^q)(\tfrac{\mathbf{1_n}}{n}-\boldsymbol{e}_i), G(x_{i_{j'}}^{j'},\xi_{*,j'}^{i_{j'}})\prod_{q=j'+1}^{t}(W_{i_q}^q-\bar{W}^q)(\tfrac{\mathbf{1_n}}{n}-\boldsymbol{e}_i)\rangle$$

$$\leq 2\mathbb{E}_{\xi\sim\mathcal{D}_i}\sum_{j=0}^{t}\sum_{j'=j+1}^{t}\left(\frac{\left\|G(x_{i_j}^j,\xi_{*,j}^{i_j})\prod_{q=j+1}^{t}(W_{i_q}^q-\bar{W}^q)(\tfrac{\mathbf{1_n}}{n}-\boldsymbol{e}_i)\right\|^2}{2}\right.$$

$$\left.+\frac{\left\|G(x_{i_{j'}}^{j'},\xi_{*,j'}^{i_{j'}})\prod_{q=j'+1}^{t}(W_{i_q}^q-\bar{W}^q)(\tfrac{\mathbf{1_n}}{n}-\boldsymbol{e}_i)\right\|^2}{2}\right) \tag{80}$$

$$\leq 2\,\mathbb{E}_{\xi\sim\mathcal{D}_i}\underbrace{\sum_{j=0}^{t}\left\|G(x_{i_j}^j,\xi_{*,j}^{i_j})\prod_{q=j+1}^{t}(W_{i_q}^q-\bar{W}^q)(\frac{\mathbf{1_n}}{n}-\boldsymbol{e}_i)\right\|^2}_{=B_3} \tag{81}$$

Once again, we can use the proof of Lemma 1 to bound this result.

**Finishing Bound of Term $B$.** Putting all terms together, we find that Term $B$ is bounded above by

$$\sum_{t=0}^{T-1}\sum_{i=1}^{n}p_i\left\|\bar{x}^{t+1}-x_i^{t+1}\right\|^2$$

$$\leq 2\gamma^2\sum_{t=0}^{T-1}\sum_{i=1}^{n}p_i\left(\frac{2(n-1)\sigma^2}{(1-\rho)Mn}+\frac{2(n-1)}{n}\sum_{j=0}^{t}\mathbb{E}_{i_j}\left\|\nabla f_{i_j}(x_{i_j}^j)\right\|^2\rho^{t-j}\right.$$

$$+\frac{4(n-1)\sqrt{\rho}\sigma^2}{Mn(1-\sqrt{\rho})^2}+\frac{2(n-1)}{n}\sum_{j=0}^{t}\mathbb{E}_{i_j}\left\|\nabla f_{i_j}(x_{i_j}^j)\right\|^2 2(t-j)\rho^{\frac{t-j}{2}}$$

$$+\frac{12(n-1)\sigma^2}{Mn}\left(\frac{1}{(1-\rho)}+\frac{128}{(1-\nu^2)}\right)$$

$$\left.+\frac{12(n-1)}{n}\sum_{j=0}^{t}\mathbb{E}_{i_j}\left\|\nabla f_{i_j}(x_{i_j}^j)\right\|^2\left(\rho^{t-j}+128\nu^{2(t-j)}\right)\right) \tag{82}$$

Simplifying results in

$$\leq 2\gamma^2\sum_{t=0}^{T-1}\sum_{i=1}^{n}p_i\left(\frac{4(n-1)\sigma^2}{Mn}\left(\frac{7}{2(1-\rho)}+\frac{\sqrt{\rho}}{(1-\sqrt{\rho})^2}+\frac{384}{(1-\nu^2)}\right)\right.$$

$$\left.+\frac{4(n-1)}{n}\sum_{j=0}^{t}\mathbb{E}_{i_j}\left\|\nabla f_{i_j}(x_{i_j}^j)\right\|^2\left(\frac{7}{2}\rho^{t-j}+(t-j)\rho^{\frac{t-j}{2}}+384\nu^{2(t-j)}\right)\right) \tag{83}$$

$$\leq\frac{8\gamma^2\sigma^2 T}{M}\left(\underbrace{\frac{n-1}{n}(\frac{7}{2(1-\rho)}+\frac{\sqrt{\rho}}{(1-\sqrt{\rho})^2}+\frac{384}{(1-\nu^2)})}_{:=\rho_\nu}\right)$$

$$+\frac{8(n-1)\gamma^2}{n}\sum_{t=0}^{T-1}\sum_{i=1}^{n}p_i\sum_{j=0}^{t}\mathbb{E}_{i_j}\left\|\nabla f_{i_j}(x_{i_j}^j)\right\|^2\left(\frac{7}{2}\rho^{t-j}+(t-j)\rho^{\frac{t-j}{2}}+384\nu^{2(t-j)}\right) \tag{84}$$

Using Lemma 5 results in

$$\leq\frac{8\gamma^2\sigma^2 T\rho_\nu}{M}+\frac{8(n-1)\gamma^2}{n}\sum_{t=0}^{T-1}\sum_{i=1}^{n}p_i\sum_{j=0}^{t}\left(2\left\|\sum_{i=1}^{n}p_i\nabla f_i(x_i^j)\right\|^2\right.$$

$$\left.+12L^2\sum_{i=1}^{n}p_i\left\|\bar{x}^j-x_i^j\right\|^2+6\zeta^2\right)\left(\frac{7}{2}\rho^{t-j}+(t-j)\rho^{\frac{t-j}{2}}+384\nu^{2(t-j)}\right) \tag{85}$$

$$\leq \frac{8\gamma^2\sigma^2 T\rho_\nu}{M} + \frac{16(n-1)\gamma^2}{n} \sum_{t=0}^{T-1}\sum_{j=0}^{t} \left( \left\| \sum_{i=1}^{n} p_i \nabla f_i(x_i^j) \right\|^2 \right.$$
$$\left. + 6L^2 \sum_{i=1}^{n} p_i \left\| \bar{x}^j - x_i^j \right\|^2 + 3\zeta^2 \right) \left( \frac{7}{2}\rho^{t-j} + (t-j)\rho^{\frac{t-j}{2}} + 384\nu^{2(t-j)} \right) \tag{86}$$

$$\leq \frac{8\gamma^2\sigma^2 T\rho_\nu}{M} + \frac{16(n-1)\gamma^2}{n} \sum_{j=0}^{T-1}\sum_{t=j+1}^{\infty} \left( \left\| \sum_{i=1}^{n} p_i \nabla f_i(x_i^j) \right\|^2 \right.$$
$$\left. + 6L^2 \sum_{i=1}^{n} p_i \left\| \bar{x}^j - x_i^j \right\|^2 + 3\zeta^2 \right) \left( \frac{7}{2}\rho^{t-j} + (t-j)\rho^{\frac{t-j}{2}} + 384\nu^{2(t-j)} \right) \tag{87}$$

$$\leq \frac{8\gamma^2\sigma^2 T\rho_\nu}{M} + \frac{16(n-1)\gamma^2}{n} \sum_{j=0}^{T-1} \left( \left\| \sum_{i=1}^{n} p_i \nabla f_i(x_i^j) \right\|^2 \right.$$
$$\left. + 6L^2 \sum_{i=1}^{n} p_i \left\| \bar{x}^j - x_i^j \right\|^2 + 3\zeta^2 \right) \left( \sum_{h=0}^{\infty} \frac{7}{2}\rho^{h} + h\rho^{\frac{h}{2}} + 384\nu^{2h} \right) \tag{88}$$

$$\leq \frac{8\gamma^2\sigma^2 T\rho_\nu}{M} + 16\rho_\nu\gamma^2 \sum_{j=0}^{T-1} \left( \left\| \sum_{i=1}^{n} p_i \nabla f_i(x_i^j) \right\|^2 + 6L^2 \sum_{i=1}^{n} p_i \left\| \bar{x}^j - x_i^j \right\|^2 + 3\zeta^2 \right) \tag{89}$$

Using Equation 38 ends with

$$\leq \frac{8\gamma^2\sigma^2 T\rho_\nu}{M} + 48T\rho_\nu\gamma^2\zeta^2 + 16\rho_\nu\gamma^2 \sum_{t=0}^{T-1} \left\| \sum_{i=1}^{n} p_i \nabla f_i(x_i^t) \right\|^2$$
$$+ 96L^2\rho_\nu\gamma^2 \sum_{t=0}^{T-1}\sum_{i=1}^{n} p_i \left\| \bar{x}^{t+1} - x_i^{t+1} \right\|^2 \tag{90}$$

Now subtract the final term on the right hand side from both sides

$$\sum_{t=0}^{T-1}\sum_{i=1}^{n} p_i \left\| \bar{x}^{t+1} - x_i^{t+1} \right\|^2 \left( \underbrace{1 - 96L^2\rho_\nu\gamma^2}_{:=z} \right) \leq \frac{8\gamma^2\sigma^2 T\rho_\nu}{M} + 48T\rho_\nu\gamma^2\zeta^2$$
$$+ 16\rho_\nu\gamma^2 \sum_{t=0}^{T-1} \left\| \sum_{i=1}^{n} p_i \nabla f_i(x_i^t) \right\|^2 \tag{91}$$

By Lemma 6, we can divide $z$ from both sides

$$\sum_{t=0}^{T-1}\sum_{i=1}^{n} p_i \left\| \bar{x}^{t+1} - x_i^{t+1} \right\|^2 \leq \frac{8\gamma^2\sigma^2 T\rho_\nu}{Mz} + \frac{48T\rho_\nu\gamma^2\zeta^2}{z} + \frac{16\rho_\nu\gamma^2}{z} \sum_{t=0}^{T-1} \left\| \sum_{i=1}^{n} p_i \nabla f_i(x_i^t) \right\|^2 \tag{92}$$

**Finishing Bound of Term $A$.** Substituting the bound of $B$ above into Equation 41 yields

$$f(\bar{x}^T) - f(\bar{x}^0) \leq -\frac{\gamma}{2n} \sum_{t=0}^{T-1} \mathbb{E}\left\| \nabla f(\bar{x}^t) \right\|^2 - \frac{\gamma}{2n} \sum_{t=0}^{T-1} \left\| \sum_{i=1}^{n} p_i \nabla f_i(x_i^t) \right\|^2$$
$$+ \left( \underbrace{\frac{3L}{(n^2-3)} + \frac{\gamma L^2 p_{max}}{2n}}_{:=\phi} \right) \left( \frac{8\gamma^2\sigma^2 T\rho_\nu}{Mz} + \frac{48T\rho_\nu\gamma^2\zeta^2}{z} \right.$$
$$\left. + \frac{16\rho_\nu\gamma^2}{z} \sum_{t=0}^{T-1} \left\| \sum_{i=1}^{n} p_i \nabla f_i(x_i^t) \right\|^2 \right) \tag{93}$$

Rearranging terms simplifies the inequality above to

$$f(\bar{x}^T) - f(\bar{x}^0) \leq -\frac{\gamma}{2n} \sum_{t=0}^{T-1} \mathbb{E}\left\| \nabla f(\bar{x}^t) \right\|^2 + \frac{\gamma}{2n} \left( \frac{32\rho_\nu\phi\gamma n}{z} - 1 \right) \sum_{t=0}^{T-1} \left\| \sum_{i=1}^{n} p_i \nabla f_i(x_i^t) \right\|^2$$
$$+ \frac{8\phi\gamma^2\sigma^2 T\rho_\nu}{Mz} + \frac{48\phi T\rho_\nu\gamma^2\zeta^2}{z} \tag{94}$$

From Lemma 8, we find that $(1 - \frac{32\rho_\nu \phi \gamma n}{z}) \geq 0$. Therefore, the second term of the right hand side above can be removed.

$$f(\bar{x}^T) - f(\bar{x}^0) \leq -\frac{\gamma}{2n} \sum_{t=0}^{T-1} \mathbb{E} \left\| \nabla f(\bar{x}^t) \right\|^2 + \frac{8\phi\gamma^2\sigma^2 T\rho_\nu}{Mz} + \frac{48\phi T\rho_\nu \gamma^2 \zeta^2}{z} \tag{95}$$

Rearranging the inequality above and dividing by $T$ yields

$$\frac{1}{T} \sum_{t=0}^{T-1} \mathbb{E} \left\| \nabla f(\bar{x}^t) \right\|^2 \leq \frac{2n\big(f(\bar{x}^0) - f(\bar{x}^T)\big)}{T\gamma} + \frac{16n\phi\gamma\sigma^2\rho_\nu}{Mz} + \frac{96n\phi\rho_\nu\gamma\zeta^2}{z} \tag{96}$$

$$= \frac{2n\big(f(\bar{x}^0) - f(\bar{x}^T)\big)}{T\gamma} + \frac{16\gamma\phi n\rho_\nu}{z}\big(\frac{\sigma^2}{M} + 6\zeta^2\big) \tag{97}$$

From Lemmas 6 and 7, the inequality above becomes

$$\frac{1}{T} \sum_{t=0}^{T-1} \mathbb{E} \left\| \nabla f(\bar{x}^t) \right\|^2 \leq \frac{2n\big(f(\bar{x}^0) - f(\bar{x}^T)\big)}{T\gamma} + \frac{\frac{1921}{10}L\gamma\rho_\nu}{n}\big(\frac{\sigma^2}{M} + 6\zeta^2\big) \tag{98}$$

Substituting in the defined step-size $\gamma$ (as well as its bound) yields

$$\frac{1}{T} \sum_{t=0}^{T-1} \mathbb{E} \left\| \nabla f(\bar{x}^t) \right\|^2 \leq \frac{2n\big(f(\bar{x}^0) - f(\bar{x}^T)\big)}{T} \left( \frac{\sqrt{TL} + \sqrt{M}}{\sqrt{Mn^2\Delta_f}} \right)$$

$$+ \frac{\frac{1921}{10}L}{n} \left( \frac{\sqrt{Mn^2\Delta_f}}{\sqrt{TL}} \right) \rho_\nu\big(\frac{\sigma^2}{M} + 6\zeta^2\big) \tag{99}$$

$$= \frac{2\sqrt{\Delta_f}}{T} + \frac{2\sqrt{L\Delta_f}}{\sqrt{TM}} + \frac{\frac{1921}{10}\sqrt{L\Delta_f}\rho_\nu\big(\sigma^2 + 6\zeta^2\sqrt{M}\big)}{\sqrt{TM}} \tag{100}$$

The final desired result is shown as

$$\frac{1}{T} \sum_{t=0}^{T-1} \mathbb{E} \left\| \nabla f(\bar{x}^t) \right\|^2 \leq \frac{2\sqrt{\Delta_f}}{T} + \frac{2\sqrt{L\Delta_f}\left(1 + \frac{1921}{20}\rho_\nu\big(\sigma^2 + 6\zeta^2\sqrt{M}\big)\right)}{\sqrt{TM}} \tag{101}$$

$\square$

# E  Additional Lemmas

**Lemma 2** (From Lemma 3 in Lian et al. 2018 (Lian et al., 2018)). *Let $W^t$ be a symmetric doubly stochastic matrix for each iteration $t$. Then*

$$\left\| \frac{\mathbf{1}_n}{n} - \prod_{t=1}^{T} W^t e_i \right\|^2 \leq \frac{n-1}{n}\rho^T, \ \forall T \geq 0.$$

**Lemma 3** (From Corollary 2 in Nedic and Olshevsky 2014 (Nedić & Olshevsky, 2014)). *Let the communication graph $\mathcal{G}$ be uniformly strongly connected (otherwise known as $B$-strongly-connected for some integer $B > 0$), and $A(t) \in \mathbb{R}^{n \times n}$ be a column stochastic matrix with $[A(t)]_{i,i} \geq 1/n \ \forall i, t$. Define the product of matrices $A(t)$ through $A(s)$ (for $t \geq s \geq 0$) as $A(t:s) := A(t)\dots A(s)$. Then, there exists a stochastic vector $\phi(t) \in \mathbb{R}^n$ such that*

$$|[A(t:s)]_{i,j} - \phi_i(t)| \leq C\nu^{t-s}$$

*will always hold for the following values of $C$ and $\nu$,*

$$C = 4, \quad \nu = (1 - 1/n^{nB})^{1/B} < 1.$$

**Lemma 4.** *Under Assumption 1, the following inequality holds*

$$\mathbb{E}_{\xi \sim \mathcal{D}_i} \left\| \frac{1}{M} \sum_{m=1}^{M} \nabla \ell(x_i^t, \xi_{m,t}^i) \right\|^2 \leq \frac{\sigma^2}{M} + \left\| \nabla f_i(x_i^t) \right\|^2.$$

**Lemma 5.** *Under Assumption 1, the following inequality holds*

$$\mathbb{E}_i \left\| \nabla f_i(x_i^t) \right\|^2 \leq 2 \left\| \sum_{i=1}^{n} p_i \nabla f_i(x_i^t) \right\|^2 + 12L^2 \sum_{i=1}^{n} p_i \left\| \bar{x}^t - x_i^t \right\|^2 + 6\zeta^2.$$

**Lemma 6.** *Given the defined step-size $\gamma$ and total iterations $T$ in Theorem 1, the term $z$ is bounded*

$$1 > z := 1 - 96L^2\rho_\nu\gamma^2 \geq 1 - \frac{384}{193^2(775)}.$$

**Lemma 7.** *Given the defined step-size $\gamma$ and total iterations $T$ in Theorem 1, the term $\phi$ is bounded*

$$\phi := \frac{3L}{(n^2 - 3)} + \frac{L^2\gamma p_{max}}{2n} \leq \frac{L}{n^2}(12 + \frac{2}{775(193)}).$$

**Lemma 8.** *Given the defined step-size $\gamma$ and total iterations $T$ in Theorem 1, we find the following bound*

$$\left(1 - \frac{32\rho_\nu\phi\gamma n}{z}\right) \geq 0$$

# F Lemma Proofs

*Proof of Lemma 1.* These steps are shown in the Main Theorem proof. Due to the symetry and double stochasticity of $\bar{W}$ we find

$$\mathbb{E}_{\xi\sim\mathcal{D}_i} \sum_{j=0}^{t} \left\| G(x_{i_j}^j, \xi_{*,j}^{i_j}) \prod_{q=j+1}^{t} (W_{i_q}^q - \bar{W}^q)(\frac{\mathbf{1_n}}{n} - \mathbf{e}_i) \right\|^2 \tag{102}$$

$$= \mathbb{E}_{\xi\sim\mathcal{D}_i} \sum_{j=0}^{t} \left\| G(x_{i_j}^j, \xi_{*,j}^{i_j}) \prod_{q=j+1}^{t} (W_{i_q}^q - \phi\mathbf{1}^\intercal + \phi\mathbf{1}^\intercal - \bar{W}^q)(\frac{\mathbf{1_n}}{n} - \mathbf{e}_i) \right\|^2 \tag{103}$$

$$= \mathbb{E}_{\xi\sim\mathcal{D}_i} \sum_{j=0}^{t} \left\| G(x_{i_j}^j, \xi_{*,j}^{i_j})\left[ \prod_{q=j+1}^{t}(W_{i_q}^q - \phi\mathbf{1}^\intercal)(\frac{\mathbf{1_n}}{n} - \mathbf{e}_i) - \prod_{q=j+1}^{t}(\bar{W}^q - \phi\mathbf{1}^\intercal)(\frac{\mathbf{1_n}}{n} - \mathbf{e}_i) \right] \right\|^2 \tag{104}$$

$$\leq 2\mathbb{E}_{\xi\sim\mathcal{D}_i} \sum_{j=0}^{t} \left\| G(x_{i_j}^j, \xi_{*,j}^{i_j}) \prod_{q=j+1}^{t}(W_{i_q}^q - \phi\mathbf{1}^\intercal)(\frac{\mathbf{1_n}}{n} - \mathbf{e}_i) \right\|^2$$

$$+ 2\mathbb{E}_{\xi\sim\mathcal{D}_i} \sum_{j=0}^{t} \left\| G(x_{i_j}^j, \xi_{*,j}^{i_j}) \prod_{q=j+1}^{t}(\bar{W}^q - \phi\mathbf{1}^\intercal)(\frac{\mathbf{1_n}}{n} - \mathbf{e}_i) \right\|^2 \tag{105}$$

Due to the structure of $\phi\mathbf{1}^\intercal$, multiplying this matrix by $\frac{\mathbf{1_n}}{n}$ or $\mathbf{e}_i$ yields the same result. Using this, as well as the double stochasticity of $\bar{W}$, we find

$$= 2\mathbb{E}_{\xi\sim\mathcal{D}_i} \sum_{j=0}^{t} \left[ \left\| G(x_{i_j}^j, \xi_{*,j}^{i_j}) \prod_{q=j+1}^{t}(W_{i_q}^q - \phi\mathbf{1}^\intercal)(\frac{\mathbf{1_n}}{n} - \mathbf{e}_i) \right\|^2 + \left\| G(x_{i_j}^j, \xi_{*,j}^{i_j}) \prod_{q=j+1}^{t}\bar{W}^q(\frac{\mathbf{1_n}}{n} - \mathbf{e}_i) \right\|^2 \right] \tag{106}$$

$$= 2\mathbb{E}_{\xi\sim\mathcal{D}_i} \sum_{j=0}^{t} \left[ \left\| G(x_{i_j}^j, \xi_{*,j}^{i_j}) \prod_{q=j+1}^{t}(W_{i_q}^q - \phi\mathbf{1}^\intercal)(\frac{\mathbf{1_n}}{n} - \mathbf{e}_i) \right\|^2 + \underbrace{\left\| G(x_{i_j}^j, \xi_{*,j}^{i_j})(\frac{\mathbf{1_n}}{n} - \prod_{q=j+1}^{t}\bar{W}^q\mathbf{e}_i) \right\|^2}_{=B_1} \right] \tag{107}$$

Using the result of Lemma 3, as our communication graph $\mathcal{G}$ is uniformly strongly connected and $[W_{i_t}^t]_{i,i} \geq 1/n$ by construction, we see that

$$|[\prod_{q=j+1}^{t} W_{i_q}^q - \phi\mathbf{1}^\intercal]_{h,k}| \leq 4\nu^{t-j-1} \,\forall\, h, k. \tag{108}$$

Using this result, we find that

$$|[\prod_{q=j+1}^{t}(W_{i_q}^q - \phi\mathbf{1}^\intercal)(\frac{\mathbf{1_n}}{n} - \mathbf{e}_i)]_h| \leq 8(\frac{n-1}{n})\nu^{t-j-1} \,\forall\, h. \tag{109}$$

We can remove the -1 exponent by doubling the constant out front

$$|[\prod_{q=j+1}^{t}(W_{i_q}^q - \phi\mathbf{1}^\intercal)(\frac{\mathbf{1_n}}{n} - \mathbf{e}_i)]_h| \leq 16(\frac{n-1}{n})\nu^{t-j} \,\forall\, h. \tag{110}$$

Since $G(x_{i_j}^j, \xi_{*,j}^{i_j})$ is all zeros except for one column, the $i_j$-th column, we find

$$\mathbb{E}_{\xi \sim \mathcal{D}_i} \sum_{j=0}^{t} \left\| G(x_{i_j}^j, \xi_{*,j}^{i_j}) \prod_{q=j+1}^{t} (W_{i_q}^q - \phi 1^\top)(\frac{\mathbf{1_n}}{n} - \boldsymbol{e}_i) \right\|^2$$

$$\leq \sum_{j=0}^{t} 256(\frac{n-1}{n})^2 \nu^{2(t-j)} \mathbb{E}_{\xi \sim \mathcal{D}_i} \left\| \frac{1}{M} \sum_{m=1}^{M} \nabla \ell(x_{i_j}^j, \xi_{*,j}^{i_j}) \right\|^2. \tag{111}$$

Utilizing Lemma 4 yields

$$\leq 256(\frac{n-1}{n})^2 \sum_{j=0}^{t} \nu^{2(t-j)} \left( \frac{\sigma^2}{M} + \left\| \nabla f_{i_j}(x_{i_j}^t) \right\|^2 \right). \tag{112}$$

By properties of geometric series, and taking the expectation over worker $i_j$, we find

$$\leq \frac{256\sigma^2}{(1-\nu^2)M}(\frac{n-1}{n})^2 + 256(\frac{n-1}{n})^2 \sum_{j=0}^{t} \mathbb{E}_{i_j} \left\| \nabla f_{i_j}(x_{i_j}^t) \right\|^2 \nu^{2(t-j)}. \tag{113}$$

Using the bound Equation 54 (term $B_1$) in the main proof above, we arrive at the final bound

$$\mathbb{E}_{\xi \sim \mathcal{D}_i} \sum_{j=0}^{t} \left\| G(x_{i_j}^j, \xi_{*,j}^{i_j}) \prod_{q=j+1}^{t} (W_{i_q}^q - \bar{W}^q)(\frac{\mathbf{1_n}}{n} - \boldsymbol{e}_i) \right\|^2$$

$$\leq \frac{512\sigma^2}{(1-\nu^2)M}(\frac{n-1}{n})^2 + \frac{4(n-1)\sigma^2}{(1-\rho)Mn} + 512(\frac{n-1}{n})^2 \sum_{j=0}^{t} \mathbb{E}_{i_j} \left\| \nabla f_{i_j}(x_{i_j}^t) \right\|^2 \nu^{2(t-j)}$$

$$+ \frac{4(n-1)}{n} \sum_{j=0}^{t} \mathbb{E}_{i_j} \left\| \nabla f_{i_j}(x_{i_j}^j) \right\|^2 \rho^{t-j} \tag{114}$$

$$\leq \frac{4(n-1)\sigma^2}{Mn} \left( \frac{1}{(1-\rho)} + \frac{128}{(1-\nu^2)} \right)$$

$$+ \frac{4(n-1)}{n} \sum_{j=0}^{t} \mathbb{E}_{i_j} \left\| \nabla f_{i_j}(x_{i_j}^j) \right\|^2 \left( \rho^{t-j} + 128\nu^{2(t-j)} \right) \tag{115}$$

Thus, we have our desired result

$$\mathbb{E} \sum_{j=0}^{t} \left\| G(x_{i_j}^j, \xi_{*,j}^{i_j}) \prod_{q=j+1}^{t} (W_{i_q}^q - \bar{W}^q)(\frac{\mathbf{1_n}}{n} - \boldsymbol{e}_i) \right\|^2 \leq \mathcal{O}(\frac{\sigma^2}{M} + \mathbb{E} \sum_{j=0}^{t} \left\| \nabla f_{i_j}(x_{i_j}^j) \right\|^2). \tag{116}$$

$\square$

*Proof of Lemma 4.*

$$\mathbb{E}_{\xi \sim \mathcal{D}_i} \left\| \frac{1}{M} \sum_{m=1}^{M} \nabla \ell(x_i^t, \xi_{m,t}^i) \right\|^2 \tag{117}$$

$$= \frac{1}{M^2} \mathbb{E}_{\xi \sim \mathcal{D}_i} \left\| \sum_{m=1}^{M} \nabla \ell(x_i^t, \xi_{m,t}^i) - \nabla f_i(x_i^t) + \nabla f_i(x_i^t) \right\|^2 \tag{118}$$

$$= \frac{1}{M^2} \mathbb{E}_{\xi \sim \mathcal{D}_i} \left\| \sum_{m=1}^{M} \nabla \ell(x_i^t, \xi_{m,t}^i) - \nabla f_i(x_i^t) \right\| + \mathbb{E}_{\xi \sim \mathcal{D}_i} \left\| \sum_{m=1}^{M} \nabla f_i(x_i^t) \right\|^2$$

$$+ \frac{2}{M^2} \mathbb{E}_{\xi \sim \mathcal{D}_i} \left\langle \sum_{m=1}^{M} \left( \nabla \ell(x_i^t, \xi_{m,t}^i) - \nabla f_i(x_i^t) \right), \sum_{m=1}^{M} \nabla f_i(x_i^t) \right\rangle \tag{119}$$

$$= \frac{1}{M^2} \sum_{m=1}^{M} \mathbb{E}_{\xi \sim \mathcal{D}_i} \left\| \nabla \ell(x_i^t, \xi_{m,t}^i) - \nabla f_i(x_i^t) \right\| + M^2 \left\| \nabla f_i(x_i^t) \right\|^2 \tag{120}$$

$$\leq \frac{\sigma^2}{M} + \left\| \nabla f_i(x_i^t) \right\|^2 \tag{121}$$

$$\square$$

*Proof of Lemma 5.*

$$\mathbb{E}_i \left\| \nabla f_i(x_i^t) \right\|^2 = \mathbb{E}_i \left\| \nabla f_i(x_i^t) - \sum_{i'=1}^{n} p_{i'} \nabla f_i(x_{i'}^t) + \sum_{i'=1}^{n} p_{i'} \nabla f_i(x_{i'}^t) \right\|^2 \tag{122}$$

$$\leq 2 \mathbb{E}_i \left( \left\| \nabla f_i(x_i^t) - \sum_{j=1}^{n} p_j \nabla f_j(x_j^t) \right\|^2 + \left\| \sum_{j=1}^{n} p_j \nabla f_j(x_j^t) \right\|^2 \right) \tag{123}$$

The first term on the right hand side can be bounded by

$$\mathbb{E}_i \left\| \nabla f_i(x_i^t) - \sum_{j=1}^{n} p_j \nabla f_j(x_j^t) \right\|^2$$

$$= 3 \mathbb{E}_i \left( \left\| \nabla f_i(x_i^t) - \nabla f_i(\frac{X^t \mathbf{1}_n}{n}) \right\|^2 + \left\| \sum_{j=1}^{n} p_j \nabla f_j(\frac{X^t \mathbf{1}_n}{n}) - \sum_{j=1}^{n} p_j \nabla f_j(x_j^t) \right\|^2 \right.$$

$$+ \left\| \nabla f_i(\frac{X^t \mathbf{1}_n}{n}) - \sum_{j=1}^{n} p_j \nabla f_j(\frac{X^t \mathbf{1}_n}{n}) \right\|^2 \right) \tag{124}$$

$$\leq 3 \mathbb{E}_i \left( L^2 \left\| x_i^t - \bar{x}^t \right\|^2 + \sum_{j=1}^{n} p_j \left\| \nabla f_j(\bar{x}^t) - \nabla f_j(x_j^t) \right\|^2 + \left\| \nabla f_i(\bar{x}^t) - \nabla f(\bar{x}^t) \right\|^2 \right) \tag{125}$$

$$\leq 3 \mathbb{E}_i \left( L^2 \left\| x_i^t - \bar{x}^t \right\|^2 + L^2 \sum_{j=1}^{n} p_j \left\| \bar{x}^t - x_j^t \right\|^2 \right) + 3\zeta^2 \tag{126}$$

$$= 6L^2 \sum_{i=1}^{n} p_i \left\| \bar{x}^t - x_i^t \right\|^2 + 3\zeta^2 \tag{127}$$

Combining all terms yields the final result

$$\mathbb{E}_i \left\| \nabla f_i(x_i^t) \right\|^2 \leq 2 \left\| \sum_{i=1}^{n} p_i \nabla f_i(x_i^t) \right\|^2 + 12L^2 \sum_{i=1}^{n} p_i \left\| \bar{x}^t - x_i^t \right\|^2 + 6\zeta^2 \tag{128}$$

$$\square$$

*Proof of Lemma 6.* It is trivial to see that $z < 1$. We now determine the lower bound of $z$ given $n \geq 2$, $p_{max} \geq 1/n$, and $\rho_\nu \geq 775/4$

$$z = 1 - 96L^2\rho_\nu\gamma^2 = 1 - 96L^2\rho_\nu(\frac{Mn^2\Delta_f}{TL}) = 1 - 96L^2\rho_\nu(\frac{1}{193^2L^2\rho_\nu^2n^2p_{max}^2}) \tag{129}$$

$$= 1 - \frac{96}{193^2\rho_\nu n^2p_{max}^2} \leq 1 - \frac{384}{193^2(775)} \tag{130}$$

$\square$

*Proof of Lemma 7.* Given $n \geq 2$ and $\rho_\nu \geq 775/4$, and the definition of $\gamma$, one can see

$$\phi = (\frac{3L}{(n^2-3)} + \frac{L^2\gamma p_{max}}{2n}) = \frac{L}{n^2}(\frac{3}{(1-3/n^2)} + \frac{L\gamma np_{max}}{2}) \tag{131}$$

$$\leq \frac{L}{n^2}(\frac{3}{(1-3/n^2)} + \frac{Lnp_{max}}{2}(\frac{\sqrt{Mn^2\Delta_f}}{\sqrt{TL}})) \tag{132}$$

Using the definition of $T$ it follows that

$$\phi \leq \frac{L}{n^2}(\frac{3}{(1-3/n^2)} + \frac{Lnp_{max}}{2}(\frac{1}{193L\rho_\nu np_{max}})) \leq \frac{L}{n^2}(\frac{3}{(1-3/4)} + \frac{1}{386\rho_\nu}) \tag{133}$$

$$= \frac{L}{n^2}(12 + \frac{2}{775(193)}) \tag{134}$$

$\square$

*Proof of Lemma 8.* Given $n \geq 2$, $p_{max} \geq 1/n$ Lemma 6, and Lemma 7, one can see

$$1 - 1 - \frac{32\rho_\nu\phi\gamma n}{z} = 1 - \frac{32\rho_\nu\phi n}{z}(\frac{1}{193L\rho_\nu np_{max}}) = 1 - \frac{32n\phi}{193Lz} \tag{135}$$

$$\geq 1 - \frac{32(12 + \frac{2}{775(193)})}{193zn} \geq 1 - \frac{16(12 + \frac{2}{775(193)})}{193(1 - \frac{384}{193^2(775)})} \geq 0 \tag{136}$$

$\square$