# OpenReview forum: "SWIFT: Rapid Decentralized Federated Learning via Wait-Free Model Communication"
_NeurIPS.cc/2022/Workshop/Federated_Learning — FL-NeurIPS 2022 Oral_

### Official Review · Reviewer_ia46 · 2022-10-17
**The paper proposes an asynchronous decentralized learning with theoretical guarantees**

This paper proposes SWIFT, an asynchronous decentralized federated learning algorithm. The authors show the proposed algorithm have convergence rate of $\frac{1}{\sqrt{T}}$, which matches the standard convergence rate for synchronous FL algorithms. Although, I do not check the proof.  In the end, authors also provide empirical evaluations over the proposes algorithm.

---

### Official Review · Reviewer_f19h · 2022-10-18
**Good paper, not as novel with respect to other commuties as the authors claim**

The authors consider a decentralized federated model with a single active client where the selection is random. The main contribution of the paper is proposing a weight matrix that is doubly-stochastic in expectation but its realization may not be. Additionally, the authors derived an analytical convergence guarantee which scales as order of 1/\sqrt{T}.

__Originality:__ The results are original with respect to the federated learning community. Nonetheless, they are less original than the authors indicated and other communities have dealt with related models and solutions.

__Pros:__

1) The paper studies an interesting and relevant model.

2) The paper proposes a practical approach for overcoming the randomness in the client selection that is mathematically sound.

3) The paper provides a thorough analysis of the convergence guarantees of their proposed algorithm.

4) The paper includes simulation results that show the accelerated convergence time that the SWIFT algorithm leads to.

5) The paper is well written.

__Cons:__

1) Points (2) and (4) are exaggerated,  many contributions such as (2) and (4)  are available in the distributed optimization community, some even date back to 2011, see for example:

       Asynchronous Broadcast-Based Convex Optimization Over a Network, Angelia Nedic ́, in IEEE Transactions on Automatic Control, VOL. 56, NO. 6, JUNE 2011

       Convergence of a Multi-Agent Projected Stochastic Gradient Algorithm for Non-Convex Optimization by Pascal Bianchi and Jérémie Jakubowicz, in IEEE Transactions on Automatic Control, VOL. 58, NO. 2, FEBRUARY 2013.

The authors should have discussed them in more detail in their related work section.

2) The SWIFT algorithm is asynchronous, however, it is still limited to a single active client. This is not ideal if the network includes a very large number of clients. I would be interested to know if the authors consider sampling a single client in each non-overlapping neighborhood.

3) The idea of utilizing a weight matrix that is doubly stochastic in expectation has been proposed in

       Asynchronous Broadcast-Based Convex Optimization Over a Network, Angelia Nedic ́, in IEEE Transactions on Automatic Control, VOL. 56, NO. 6, JUNE 2011

for distributed optimization, the authors should point this out.

4) Generally, terms such as novel, first, etc. often proved to be incorrect, please refrain from using them unless absolutely necessary. As you can find from my earlier comments, the idea is not as novel as thought to be and was already discussed in a related community.

5) The authors do not consider random link failure between clients.

6) "In synchronous algorithms, the global iteration t increases only after all clients finish an update." I suggest you accompany this sentence with a relevant citation.

7) Figure 9 is a replicate from another paper, while the authors clearly stated that it is best not to copy a figure from another paper since it can be intellectual property. The best practice is to generate the figure, note that and cite the original figure.

In spite of these weak points which I included for the benefit of the anonymous authors, the paper is very good.
It will be a very nice addition to the FL theory.

---

### Official Review · Reviewer_gc7b · 2022-10-19
**The paper proposes a new technique for decentralized FL with asynchronous communication between clients**

Overall the contribution of the paper is good. The authors consider the problem of decentralized federated learning where the central server is not required and implement a distributed communication strategy between clients. The algorithm proposes an asynchronous strategy where clients can run gradient updates at their own speed. The paper provided good theoretical guarantees and numerical experiments to save communication costs.

---

### Decision · Program_Chairs · 2022-10-20

Accept (Oral)